# DISTILL NOT ONLY DATA BUT ALSO REWARDS: CAN SMALLER LANGUAGE MODELS SURPASS LARGER ONES?

## ABSTRACT

Distillation of large language models (LLMs) has traditionally focused on transferring teacher responses, often assuming access to internal logits. In modern LLM deployment, however, the teacher is typically only accessible as a black-box API or is too large to support online distillation, while simultaneously possessing strong evaluative capabilities that remain underexploited. As a result, students learn what to answer, but not which answers are preferable. This gap limits generalization, propagates teacher errors, and prevents students from improving beyond imitation. Therefore, we propose a unified distillation framework that transfers both responses and evaluation ability. Our key idea is to distill reward signals from the teacher, eliminating the need for costly human annotations. However, extracting reliable reward signals from LLMs is challenging because they are optimized for generation rather than evaluation. Therefore, we introduce an adaptive reward distillation strategy that applies majority voting for verifiable tasks and LLM-as-Judge for open-ended tasks. This yields noisy yet effective self-supervised signals without human annotations. To mitigate distribution shift, we systematically collect and label both teacher- and student-generated responses, which are used to train a reward model. The student is first warmed up with supervised fine-tuning on high-quality teacher responses, then refined with reinforcement learning guided by the learned reward model. Experiments on GSM8K, GSM-Plus, MMLU-Pro, and AlpacaEval2 demonstrate consistent gains over supervised fine-tuning, with smaller students in some cases even surpassing their teachers. These results highlight our method as a scalable and effective paradigm for training efficient yet competitive LLMs. Code: Link.

## 1 INTRODUCTION

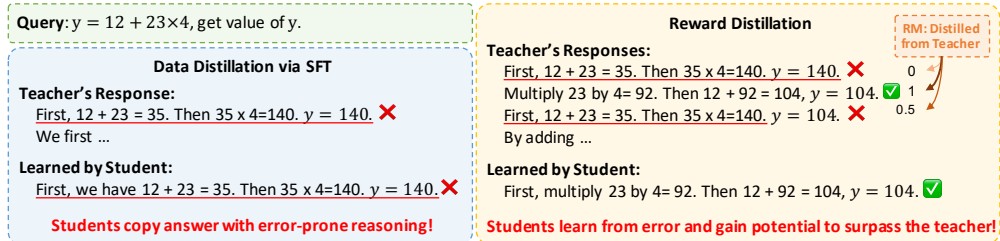

Figure 1: Comparison of (a) traditional data distillation and (b) reward distillation. Instead of supervised fine-tuning (SFT) on the teacher's response, reward distillation distills the teacher's evaluation into a reward model (RM), enabling the student to learn from both success and errors, thus gaining potential to surpass the teacher through reinforcement rearning.

Distillation has emerged as a promising technique for mitigating the high computational demands of large language models (LLMs) by training smaller student models under the guidance of larger teachers. However, achieving competitive performance through fine-tuning is critically dependent on high-quality annotated data, a resource that remains a significant bottleneck (Achiam et al., 2023; Kaplan et al., 2020; Roziere et al., 2023; Yuan et al., 2023; Luo et al., 2023). Additionally, in modern

deployment settings, teacher LLMs are often only accessible as black-box APIs or expensive online calling, which effectively precludes access to internal logits. Therefore, supervised fine-tuning (SFT) using distilled data has become the dominant approach (Feng et al., 2021) (*Data Distillation*), where teacher LLMs generate responses for various tasks and domains, sometimes in conjunction with search and selection strategies (Tian et al., 2024; Zhang et al., 2024).

While this *data distillation* strategy has proven effective (Magister et al., 2023; Fu et al., 2023a), it is inherently limited: it transfers *what* the teacher says but not *which answers are preferable*. As a result, such distillation propagates teacher errors without discrimination, and students are prevented from improving beyond imitation. Yet large teacher models always possess evaluative ability to judge the correctness or relative quality of responses, a dimension that current distillation pipelines largely ignore. In parallel, recent advances in reinforcement learning (RL) with human or AI feedback demonstrate the value of reward models in guiding training (Wang et al., 2024c; Setlur et al., 2024; Wang et al., 2024a), but these approaches require costly human annotation or depend on separately trained reward systems. This raises a natural research question: *can we directly distill both responses and evaluative ability from teachers themselves, without external supervision, and use these signals to train smaller students that rival or even surpass their teachers?*

Directly prompting LLMs for evaluations, however, can be biased or inconsistent, since they are primarily optimized for generation and are prone to hallucination and uncertainty issues (Huang et al., 2023; Xiong et al., 2023). Nevertheless, our empirical analysis (Table 1) shows that even noisy reward signals distilled from a teacher can substantially benefit student training. For example, distilling from a Llama3-8B teacher to a Llama3-1B student on MMLU-Pro yields clear gains, demonstrating that reward distillation is a viable and powerful way of transferring knowledge beyond response imitation.

Table 1: Distillation Performance on MMLU-Pro under different distillation strategies.

| Model | MMLU-Pro (%) |
|---|---|
| Teacher (8B) | 39.88 |
| Student (1B) | 15.97 |
| Reward Distill. | 26.44 (+10.47) |
| Data Distill. | 22.62 (+6.65) |

Motivated by this observation, we propose a uniform distillation framework that integrates both data and reward distillation. Our framework fully leverages teachers' capacity for generation and evaluation, eliminating human-labeled rewards. For verifiable tasks shown in Table 2, we derive pseudo-labels through the inherent structure of teacher and student responses and score the responses accordingly. For open-ended tasks, where the answer is non-structured and open-ended, we adopt LLM-as-Judge as the teacher's evaluation paradigm. To mitigate distribution shift, we consider both teacher- and student-generated responses when constructing the reward model training data. By forcing reward models to distinguish between correct and incorrect answers and assign higher scores to the teachers' reasoning rather than the students', we enable the reward model to adaptively capture teachers' advantages and students' weaknesses, without introducing a huge distribution shift. Finally, the student, which is first warmed up with SFT on high-quality teacher responses, is refined with RL guided by the learned reward model. This reward-driven refinement process benefits students from both response-level supervision and implicit preference signals, and in some cases, even makes student surpass their teachers, as shown by the blue marks in Table 3.

Our key contributions are threefold: (1) We design a novel distillation framework that transfers not only data but also rewards, eliminating the need for external human annotations. (2) We propose tailored strategies for generating reward signals: majority voting and structured extraction for verifiable tasks, and LLM-as-Judge for open-ended tasks. (3) Across verifiable benchmarks (GSM8K, GSM-Plus, MMLU-Pro) and open-ended evaluation (AlpacaEval2), our approach consistently outperforms SFT-based methods, with smaller students in some cases surpassing their teachers.

## 2 RELATED WORK

In this section, we review several relevant topics of our method, including distillation and reinforcement learning from external feedback.

**Distillation.** Recent studies on distillation for language models have primarily focused on transferring reasoning capabilities from large language models (LLMs) to smaller models (Shridhar et al., 2022; Magister et al., 2023). When the teacher's internal logits are accessible, soft distillation methods such as GKD (Agarwal et al., 2024) and MiniKD (Gu et al.) employ KL-style objectives to better mimic the teacher's output distribution. However, such logit-based supervision is often infeasible in

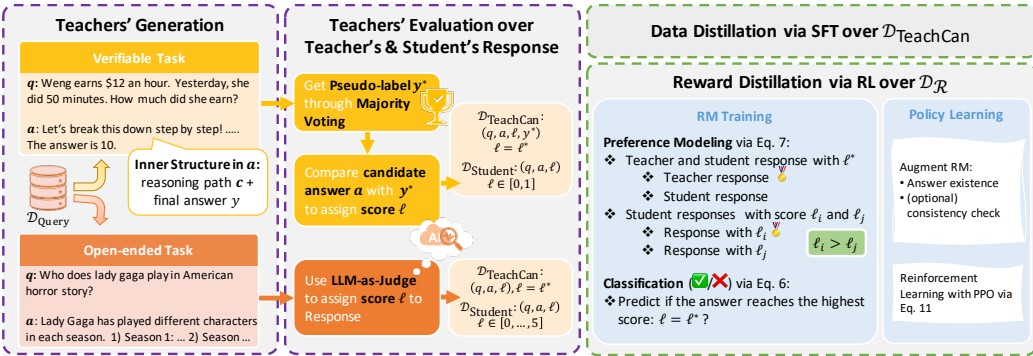

Figure 2: Overall framework. **Teacher's Generation & Evaluation:** Teachers generate answers and score both teacher and student responses using a majority vote or an LLM-as-Judge approach. $\ell$ denotes the score and $\ell^*$ is its max value. **Data Distillation:** The student is improved by SFT on high-quality teacher answers. **Reward Distillation:** A reward model is trained on these scored data ($\mathcal{D}_\mathcal{R}$, constructed from $\mathcal{D}_{\text{TeachCan}}$ and $\mathcal{D}_{\text{Student}}$) using preference and classification losses. The learned reward model then helps to further improve students through RL with answer-existence and consistency checks as augmentation.

Table 2: Example of verifiable task. The final answer is shown in blue.

| Q | Weng earns \$12 an hour for babysitting. Yesterday, she just did 50 minutes. How much did she earn? |
|---|---|
| A | Let's break this down step by step! Step 1: 50 minutes is equal to $\frac{50}{60} = \frac{5}{6}$ of an hour. Step 2: Multiply her hourly rate by the fraction of an hour she worked: $\$12 \times \frac{5}{6} = \$10$. The answer is $\$10$. |

practice due to the inaccessibility of internal outputs for commercial LLMs, the high cost of online inference with very large teachers, and potential misalignment between teacher and student tokenizers (Gou et al., 2021; Wang et al., 2025). This motivates the use of *data distillation*, which relies only on the teacher's text responses and is broadly applicable. These approaches rely on supervised fine-tuning (SFT) (Ho et al., 2023; Magister et al., 2023; Fu et al., 2023b) and leverages advanced LLMs such as the GPT series (Radford et al., 2019; Brown et al., 2020; Achiam et al., 2023) as the guidance to generate high-quality data (Josifoski et al., 2023; Li et al., 2023c). Symbolic Chain-of-Thought Distillation (Li et al., 2023a) introduced a step-by-step reasoning framework, highlighting the potential of distilling complex reasoning processes, while FLD (Morishita et al., 2023) focused on logical deductive reasoning. More recently, Guo et al. (2025) proposed distilling multiple small models via SFT over reasoning data generated by DeepSeek-R1. However, most methods treat LLMs merely as sources of reasoning chains, optimizing student models exclusively through supervised fine-tuning, ignoring the teacher LLMs' evaluation capability. Additionally, some works have explored reinforcement learning to further enhance reasoning capabilities; for instance, MARIO (Ramnath et al., 2023) employs multiple external reward models to improve self-rationalization. In contrast, our work takes a different approach by leveraging LLMs not only for response generation but also for extracting reward signals, enabling a more comprehensive distillation process.

**Reinforcement learning from External Feedback.** Following the success of Reinforcement Learning from Human Feedback (RLHF) (Ziegler et al., 2019) in enabling the widespread application of LLMs, researchers have increasingly explored ways to reduce human involvement in training through Reinforcement Learning from AI Feedback (RLAIF). As a pioneer in RLAIF, Constitutional AI (Bai et al., 2022) has demonstrated improved performance in tasks such as summarization, helpful dialogue generation, and harmless dialogue generation. Lee et al. (2023) further show that RLAIF can achieve performance comparable to or even surpassing RLHF, as evaluated by human judges. SPIN (Chen et al., 2024) eliminates the need for explicit reward models by adopting an iterative DPO-like framework, where human-labeled winning responses are paired with the previous iteration's generations as losing responses. However, those methods do not focus on training smaller models through distillation and require external reward models or human labels for optimization. Beyond the above, recent work studies more autonomous feedback generation through the language model itself (Yuan et al., 2024; Wu et al., 2025), however, those methods require the model to be strong enough to provide reliable reward signals for its self-evolvement, while we focus on the teacher-student setting where the student is significantly weaker and relies on a stronger external teacher to provide both data and evaluative signals.

## 3   PRELIMINARY

For better presentation of our method, we first introduce the preliminaries on verifiable tasks, open-ended tasks, and majority voting.

**Verifiable *v.s.* Open-ended Tasks.** We categorize the LLM tasks into verifiable tasks and open-ended tasks and thus tailor different evaluation strategies according to their characteristics. Verifiable tasks, as illustrated in Table 2, are featured by the inner structures in which each response $a$ contains a final answer $y$ and a reasoning path $c$: $y = \text{Extract}(a)$. Here, $\text{Extract}(a)$ is typically a function provided by the task to extract the final answer $y$ from $a$. For the math reasoning example in Table 2, $\text{Extract}(a)$ identifies the numerical value indicated in the language-based response. By contrast, for the open-ended tasks, the responses are generally non-structured and open-ended.

**Majority Voting** is a commonly used method in test-time optimization for verifiable tasks. Let $q$ be a query and $\{a_1, \ldots, a_m\}$ be $m$ i.i.d. sampled responses from an LLM $\mathcal{G}(\cdot \mid q)$. Define $y_i = \text{Extract}(a_i)$ for $i = 1, \ldots, m$ and the empirical vote distribution over candidate final answers, $y \in \mathcal{Y}$, by $p_q(y) = \frac{1}{m}\sum_{i=1}^{m} \mathbb{1}[y_i = y]$, where $\mathbb{1}$ is the indicator function. Then we have

$$y_q^* = \arg\max_{y \in \mathcal{Y}} p_q(y), \qquad p_q^* = \frac{1}{m}\sum_{i=1}^{m} \mathbb{1}(y_i = y^*). \tag{1}$$

Typically, a confidence threshold $sh \in (0, 1)$, is used to abstain the low-consensus final answer $y_q^*$ if $p_q^* < sh$. In general, larger $m$ and higher $sh$ yield a more reliable final answer.

## 4   METHODOLOGY

In this section, we present our unified framework (as illustrated in Figure 2) for enhancing student models by distilling both responses and reward signals from teacher LLMs. Formally, distillation starts with a large teacher model ($\mathcal{T}$), a smaller student model ($\mathcal{S}$), and a dataset ($\mathcal{D}_{\text{query}}$) containing **only** queries $q$.

### 4.1   TEACHER'S GENERATION & EVALUATION

In this subsection, we introduce response generation for data distillation (shared with traditional SFT distillation) and evaluation generation for reward distillation, which are the foundation of our framework.

**Teacher's Generation.** Similarly to traditional distillation, for each query $q$, teacher $\mathcal{T}$ generates multiple diverse responses $a$ with varied reasoning paths under high temperature. We denote the resulting datasets as $\mathcal{D}_{\mathcal{T}} := \{(q, a_i) : \forall q \in \mathcal{D}_{\text{query}}, i \in [N]\}$ where $N$ denotes the number of generations for each query.

**Adaptive Teacher's Evaluation.** Extracting reliable rewards from LLMs is hard because they are tuned for generation rather than evaluation. Therefore, we adopt an adaptive scheme, employing various evaluation strategies tailored to each task: majority voting for verifiable tasks and LLM-as-Judge for open-ended ones. Below, we use an example of a candidate response $a$ and its corresponding query $q$ to illustrate how to properly stimulate the teacher's evaluation.

*Verifiable tasks.* Leveraging the inherent structure of verifiable answers, we first adopt majority voting to derive a pseudo-label $y_q^*$ and its corresponding probability $p_q^*$ for each $q$, as illustrated in Sec. 3. In practice, we set $m = 10$, $sh = 0.7$. As a consequence, we can filter out the questions that the teacher is unable to answer, *i.e.*, dropping $q$ such that $p_q(y^*) < sh$. By aligning the extracted final answer $y$ with $y^*$, we can assign a binary score $\ell \in [0, 1]$ for each candidate response $a$. Denote the max value of $\ell$ as $\ell^*$, therefore, for verfiable task, $\ell^* = 1$.

*Open-ended tasks.* When no verifiable answer exists, we adopt the LLM-as-Judge paradigm: the teacher is prompted to assign a score according to the response quality. We include a detailed scoring scheme in the LLM-as-Judge prompt with a discrete scale from 0 to 5, *i.e.*, $\ell \in [0, 5]$ where $\ell^* = 5$ denotes the highest quality.

Note that the teacher's evaluations are always noisy, whether majority voting or LLM-as-Judge is used, AI's feedback inevitably introduces false positive labeling, as illustrated in our experiments. We list all the prompts used above for calling the teachers in Appendix B.

## 4.2 DATA DISTILLATION

We slightly warm up the student through data distillation, thereby reducing meaningless exploration during the initial RL stage. For training, we collect the query and response pairs with the highest score, forming the dataset $\mathcal{D}_{\text{TeachCan}}$:

$$\mathcal{D}_{\text{TeachCan}} = \{(q, a_i, \ell_i) : (q, a_i) \in \mathcal{D}_{\mathcal{T}}, \ell = \ell^*, i \in [N]\}, \tag{2}$$

where $\ell_i$ is the teacher-assigned score for $a_i$. The resulting dataset contains only queries that the teacher is able to answer, ensuring that the student is learned on reliably solvable instances. The training objective of this stage is to minimize the negative log-likelihood loss:

$$\mathcal{L}(\theta) = -\mathbb{E}_{(q,a)\sim\mathcal{D}_{\text{TeachCan}}} \log P(a_k | a_{<k}, q, \theta), \tag{3}$$

where $k$ denotes the index of tokens in the response $a$, and $\theta$ represents the parameters of the student model. Finally, we obtain a student model $\mathcal{S}_{\text{DataDistill}}$.

## 4.3 REWARD DISTILLATION I: REWARD MODEL LEARNING

As the core of our method, this section illustrates why and how we distill a powerful reward model from teachers' noisy evaluations. We answer two questions in this subsection:

- Whose response do we need, Teacher *v.s.* Student?
- How can we inject teacher guidance while avoiding a large distribution shift?

**Whose response do we need: Both Teacher and Student.** For the first question, training a reward model solely to separate correct from incorrect *teacher* answers is straightforward, but it induces a distribution shift because the teacher and the student exhibit different generation patterns. Conversely, relying only on student responses risks ignoring the insights provided by the teacher's superior reasoning paths. For example, the teacher's correct reasoning path can assist in correcting the students' responses with incorrect reasoning, but a correct final answer. We therefore adopt a scheme that systematically collects, filters, and labels both *teacher-* and *student-generated* responses to construct the reward training set, aligning supervision with the student's data distribution.

*Collecting labeled Student Generations.* Apart from collecting teacher responses in data distillation, we collected students' responses as supplementary to reward model training data. For each query $q \in \mathcal{D}_{\text{TeachCan}}$, we sample diverse responses with a temperature of $0.7$, and then the teacher evaluates and labels these responses, following the procedure described in Sec. 4.2. Finally, we obtain a dataset consisting of students' responses and their labels:

$$\mathcal{D}_{\text{Student}} = \{(q, a_i, \ell_i) : (q, a_i) \in \mathcal{D}_{\text{Query}}, i \in [N]\}, \tag{4}$$

where $\ell_i = \mathbb{K}(\text{Extract}(a_i) = y_q^*)$ for verifiable tasks ($\ell_i \in [0, 1]$) and LLM-as-Judge-assigned score $\ell_i \in [0, 5]$ for open-ended tasks. Notably, to train with balanced samples, we discard questions such that the student model consistently produces either only correct or only incorrect answers.

**Injection teacher guidance while avoiding a large distribution shift.** For the second question, we require the reward model to do more than separate correct from incorrect answers; *it should also assign higher scores to superior reasoning, particularly that exhibited by the teacher.* This is motivated by the observation that stronger teachers typically produce more concise and reliable reasoning than students, even when both reach the correct answer. By structuring training to distinguish responses by correctness and additionally rank then by reasoning quality, the reward model learns from diverse outputs to both detect correctness and preferentially reward more accurate reasoning. Below we illustrate how we achieve these by constructing balanced pair-wise data and training the reward model with classification and preference modeling.

*Constructing Pair-wise Training Data.* We define the following response set for query $q$,

- $\mathcal{S}(q) \subseteq \mathcal{D}_{\text{Student}}$: all student responses for query $q$;
- $\mathcal{S}^+(q) \subseteq \mathcal{D}_{\text{Student}}$: student responsesreaching the highest score *i.e.*, for $a$ with $\ell, \ell = \ell^*$;
- $\mathcal{T}^+(q) \subseteq \mathcal{D}_{\text{TeachCan}}$: teacher responses such that for $a$ with $\ell, \ell = \ell^*$.

We then construct preference pairs by comparing scores for each query $q$:

$$\mathcal{P}(q) = \{(a_i, a_j) \mid a_i \in \mathcal{T}^+, a_j \in \mathcal{S}^+\} \cup \{(a_i, a_j) \mid (a_i, a_j) \in \mathcal{S}, \ell_i > \ell_j\}, \tag{5}$$

where $\ell_i$ and $\ell_j$ denote the teacher-assigned score for $a_i$ and $a_j$, separately. We provide an example of pairwise data in Appendix G. Finally, the reward-model training set is defined as

$$\mathcal{D}_{\mathcal{R}} = \{(q, (a_i, \ell_i), (a_j, \ell_j)) \mid q \in \mathcal{D}_{\mathcal{S}}, (a_i, a_j) \in \mathcal{P}(q)\}. \tag{6}$$

Formally, our reward model, which is initialized from the student $\mathcal{S}_{\text{DataDistill}}$ takes a question–answer pair $(q, a)$ as input and outputs a scalar reward $r$.

As we mentioned, we not only optimize the reward model to distinguish the correct and incorrect answers but also force it to prefer teachers' reasoning. Guided by this, we tailored our reward model training objective to comprise two components: a classification loss and a preference-based loss, which is presented below.

1) *Classification Loss*: reward models are trained to distinguish if the answer deserves the highest score ($\ell = 1$ for verifiable tasks and $\ell = 5$ for open-ended tasks), therefore, the classification loss is,

$$\mathcal{L}_{\text{cls}}(\mathcal{R}) = -\mathbb{E}_{(q,a,\ell)\sim\mathcal{D}_{\mathcal{R}}}\left[\mathbb{1}(\ell = \ell^*)\log\sigma(\mathcal{R}(q,a)) + \mathbb{1}(\ell \neq \ell^*)\log(1 - \sigma(\mathcal{R}(q,a)))\right], \quad (7)$$

where $\sigma$ is the sigmoid function and $\ell \in [0, 1]$ denotes the labels derived by teacher's evaluation. $\ell^*$ denotes the highest score of $\ell$, which is 1 for verifiable task and 5 for open-ended task.

2) *Preference-based loss:* reward models are trained to i) assign a higher score to the teacher's response in $\mathcal{T}^+(q)$, compared with the student's correct answer in $\mathcal{S}^+(q)$; ii) a higher score to the students' better answers, compared with the worse ones (lower value for $\ell$). With the pair-wised data, the preference loss is defined as,

$$\mathcal{L}_{\text{pref}}(\mathcal{R}) = -\mathbb{E}_{q,\,a_i,\,a_j\sim\mathcal{D}_{\mathcal{R}}}\left[\log\sigma\big(\mathcal{R}(q,a_i) - \mathcal{R}(q,a_j)\big)\right], \quad (8)$$

Finally, the overall objective for reward model training is defined as,

$$\mathcal{L} = \mathcal{L}_{\text{pref}} + \lambda\mathcal{L}_{\text{cls}}. \quad (9)$$

Note that, with the preference pair datasets, we also experimented with Direct Preference Optimization (Rafailov et al., 2023) to optimize the student LLM over preference-based pairs, but found that it brings limited improvement to student performance during warm-up, as shown in Appendix E.6. Therefore, we focus on improving the warm-up model through RL training.

### 4.4 Reward Distillation II: Optimization through Reinforcement Learning

With the trained reward model $\mathcal{R}$, we can continue to optimize the student model after data distillation $\mathcal{S}_{\text{DataDistill}}$. The training data for RL is the queries drawn from $\mathcal{D}_{\mathcal{R}}$, consisting of queries that the student may fail to address.

**Reward Design.** The quality of the reward model has a significant impact on the optimization process and performance of PPO. To further augment our trained reward model without additional computational cost, we define the reward signals for training as following:

1. Reward Model Score: We use the predicted reward from the trained reward model $\mathcal{R}$. Ideally, a positive reward value indicates correctness, while a higher reward reflects a better response.
2. Answer Existence: If the model fails to provide a valid answer (or extractable answer for the verifiable tasks), we assign a reward of $-5$ and terminate the evaluation for this query.
3. Consistency Check with the Pseudo Label in $\mathcal{D}_{\mathcal{R}}$ (Only for Verifiable Task): Finally, we compare the extracted answer with the pseudo-label $y_q^*$, if applicable. If the extracted value does not match the pseudo-label, we adjust the reward using $\min(r, 0)$.

Therefore, the final reward for a response $a$ for question $q$ is defined as,

- Verifiable tasks:

$$\widetilde{\mathcal{R}}(q,a) = \begin{cases} -5, & \text{no extractable answer;} \\ \min(r,0), & \text{if } \hat{y} \neq y^*; \\ r, & \text{if } \hat{y} = y^*, \end{cases} \quad (10)$$

- Open-ended tasks:

$$\widetilde{\mathcal{R}}(q,a) = \begin{cases} -5, & \text{no valid answer;} \\ r, & \text{otherwise,} \end{cases} \quad (11)$$

where $r = \mathcal{R}(q, a)$ is the predicted reward from the reward model. We use the augmented $\tilde{\mathcal{R}}(q, a)$ in further reinforcement learning. This design ensures that the student model is discouraged from producing incomplete responses while also allowing the reward model's output to be further refined by aligning it with the pseudo-label when applicable.

**Optimization with PPO.** We employ Proximal Policy Optimization (PPO) (Schulman et al., 2017) in RL to refine the student model $\mathcal{S}_{\text{DataDistill}}$ under the supervision of reward signals defined in Eq. 10 and Eq. 11. Let $\theta$ denote the parameters of $\mathcal{S}_{\text{DataDistill}}$. The loss function for optimizing $\theta$ is,

$$\begin{aligned} \mathcal{L}_{\text{PPO}} = \quad & \mathbb{E}_{(q,a)\sim\mathcal{D}_{\mathcal{R}}\cup\mathcal{D}_{\text{fail}}}\left[\min(m_t(\theta)\hat{A}_t, \text{clip}(m_t(\theta), 1-\epsilon, 1+\epsilon)\hat{A}_t)\right] \\ & -\lambda\,\mathbb{E}\left[\text{KL}(\pi_\theta(\cdot \mid a_{<t};q) \,\|\, \pi_{\theta_{\text{old}}}(\cdot \mid a_{<t};q))\right] \end{aligned} \quad (12)$$

where $m_t(\theta) = \frac{\pi_\theta(a_t|a_{<t},q)}{\pi_{\theta_{\mathrm{old}}}(a_t|a_{<t};q)}$ represents the probability ratio between the updated policy $\pi_\theta$ and the previous policy $\pi_{\theta_{\mathrm{old}}}$. The term $\hat{A}_t$ is the estimated advantage function at time step $t$, while $\epsilon$ is the clipping threshold that constrains policy updates. The coefficient $\lambda$ controls the penalty term, and $\mathrm{KL}\left[\pi_\theta(\cdot|a_{<t};q) \,\middle\|\, \pi_{\theta_{\mathrm{old}}}(\cdot|a_{<t};q)\right]$ is the Kullback-Leibler (KL) divergence between the current and previous policies, ensuring stable updates.

## 5 EXPERIMENTS

In this section, we evaluate the efficacy of our proposed method and conduct comprehensive ablation studies to assess its effectiveness and justification.

### 5.1 SETUP

**Verifiable tasks.** We evaluate our method on three widely used benchmarks: GSM8K (Cobbe et al., 2021): A dataset of grade-school math problems designed to assess models' problem-solving and reasoning capabilities; GSM-Plus (Li et al., 2024): A variant of GSM8K that introduces various mathematical perturbations to test generalization. We train student models on GSM8K and directly evaluate them on GSM-Plus; MMLU-Pro (Wang et al., 2024b): A professional-level multi-task benchmark covering a wide range of knowledge domains, including humanities, sciences, and engineering. **Data Split:** Details of the dataset splits are provided in Appendix C.1. During training, we select the optimal checkpoints based on the student model's performance on the validation set. **Metrics.** We report accuracy as the primary evaluation metric. To ensure reproducibility, we set the generation temperature to 0 during inference.

**Open-ended tasks.** We train the student on a subset of UltraFeedback ($\approx$ 13k instructions). Ultra-Feedback contains diverse 250k user-assistant conversations from various aspects, and we sample the instructions with GPT-4 as an evaluator for our training dataset. For evaluation, we use AlpacaEval2 (Li et al., 2023b; Dubois et al., 2024; 2023), an LLM-judged benchmark, where a GPT-4-Turbo judge compares our model against a fixed baseline per prompt and produces a preference probability. **Metrics.** We report the standard win rate and the length-controlled win rate following the official AlpacaEval2 protocol.

**Evaluated Methods.** To assess the efficacy of our approach, we evaluate the following baselines and ablation variants: 1) **In-Context Learning (ICL)**: The student model utilizes in-context learning without any fine-tuning. 2) **Supervised Fine-Tuning (SFT):** The student model is trained via Supervised Fine-Tuning (SFT) on the teacher's responses. 3) **Teacher LLM:** We report the performance of teacher LLMs (Llama3-70B or Llama3-8B) to evaluate whether the student can outperform the teacher. 4) **Ours:** Our full method. The student model is trained with reinforcement learning (RL) following a warm-up phase using the teacher's responses. 5) **Ours w/o Data (Ours w/o D):** Ablation version where the student model is trained using RL in the same manner as **Ours**, but without the SFT warm-up phase utilizing the teacher's responses. 6) **Ours w/o Reward (Ours w/o R):** Ablation version where the student model is trained via SFT on the teacher's responses, with the teacher's self-evaluation employed to filter high-quality responses for training. For more implementation details, please refer to Appendix C.

### 5.2 MAIN RESULTS ON VERIFIABLE TASKS

In this subsection, we report the performance of teacher and student models of different sizes on three benchmark tasks. The results are summarized in Table 3. Notably, for GSM-Plus, the models are trained on GSM8K and evaluated on GSM-Plus, showcasing their ability to generalize beyond the training distribution.

**Our method consistently outperforms other approaches across varying teacher capacities.** With a stronger teacher, LLaMA3-70B, the 1B student achieves notable gains: from 61.03% to 64.06% on GSM8K (+3.03%) and from 31.00% to 35.27% on MMLU-Pro (+4.27%). The 3B student also benefits, with improvements of +1.90% on GSM8K and +2.96% on MMLU-Pro. When using the less capable LLaMA3-8B as the teacher, the improvements are even more pronounced. The 1B student improves by +3.18% on GSM8K and achieves a substantial +9.38% gain on MMLU-Pro (22.62% $\rightarrow$ 32.00%). Similarly, the 3B student sees increases of +3.27% on GSM8K and +8.24% on

Table 3: Evaluation across different teachers, students and datasets. The best scores are boldfaced, and the scores where the student model outperforms the teacher are marked in blue.

| Teacher | Student | Method | GSM8K | GSM-Plus | MMLU-Pro |
|---|---|---|---|---|---|
| | | Llama3-70B (Teacher) | 93.18% | 83.24% | 56.85% |
| Llama3-70B | Llama3-1B | ICL | 41.55% | 28.92% | 15.97% |
| | | SFT | 61.03% | 39.95% | 31.00% |
| | | Ours w/o R | 61.41% | 40.38% | 32.83% |
| | | Ours w/o D | 61.25 % | **42.86 %** | 26.16% |
| | | Ours | **64.06%** | 41.86% | **35.27%** |
| | Llama3-3B | ICL | 72.48% | 45.48% | 29.62% |
| | | SFT | 80.74% | 61.76% | 41.54% |
| | | Ours w/o R | 80.89% | 62.24% | 42.10% |
| | | Ours w/o D | 80.29 % | **63.76%** | 35.30% |
| | | Ours | **82.64%** | 63.05% | **44.50%** |
| | | Llama3-8B (Teacher) | 80.97% | 68.19% | 39.88% |
| Llama3-8B | Llama3-1B | ICL | 41.55% | 28.92% | 15.97% |
| | | SFT | 62.85% | 42.23% | 22.62% |
| | | Ours w/o R | 63.08% | **43.14%** | 23.85% |
| | | Ours w/o D | 60.12% | 40.95% | 26.44% |
| | | Ours | **66.03%** | 43.00% | **32.00%** |
| | Llama3-3B | ICL | 72.48% | 45.48% | 29.62% |
| | | SFT | 79.75% | 62.85% | 31.78% |
| | | Ours w/o R | 79.91% | 62.86% | 34.50% |
| | | Ours w/o D | 80.36% | 64.19% | 36.08% |
| | | Ours | **83.02%** | **64.24%** | **40.02%** |

MMLU-Pro (31.78% → 40.02%). These results demonstrate the robustness of our method, which consistently enhances student performance across different model scales and teacher qualities.

**Our method outperforms distilling the data/reward-only methods** by jointly distilling both components. We further conduct ablations to isolate the contributions of reward and data distillation. **Ours w/o Data** removes the initial supervised warm-up, while **Ours w/o Reward** retains only the filtered teacher answers for SFT without reinforcement learning. As shown in Table 3, both variants outperform the no-distillation baseline, indicating that either data or reward alone is beneficial. However, by jointly distilling both components, our method consistently yields the best results over GSM8K and MMLU-Pro.

Importantly, in some settings (in blue) **our student models even surpass their teachers**. For instance, under LLaMA3-8B supervision, the 3B student outperforms the teacher on GSM8K (83.02% vs. 80.97%) and MMLU-Pro (40.02% vs. 39.88%), illustrating the effectiveness of our combined distillation strategy. Moreover, our method provides substantial improvements even under cross-task generalization, such as from GSM8K to GSM-Plus.

## 5.3 MAIN RESULT ON OPEN-ENDED TASKS

We evaluate our method on the open-ended benchmark AlpacaEval2: train a student model, Llama3-1B, under the guidance of teacher Llama3-70B. As shown in Table 4, our method provides students with an improvement of +5.38% in win rate and +4.07 in length-controlled win rate (LC Win Rate), surpassing the traditional SFT distillation. Therefore, **our method surpasses the baselines, indicating stronger open-ended generation.**

Table 4: Open-ended task evaluation.

| Model | Win Rate | LC Win Rate |
|---|---|---|
| Teacher | 35.88 | 37.55 |
| ICL | 8.05 | 6.95 |
| SFT | 10.23 (+2.18) | 8.99 (+2.04) |
| **Ours** | **13.43** (+5.38) | **11.02** (+4.07) |

## 5.4 HOW TO OBTAIN GOOD REWARD SIGNALS

To further analyze how to obtain a good reward signal for training students, we conduct ablation studies to answer several questions. Unless otherwise specified, these ablations are performed using the teacher model Llama3-70B, the student model Llama3-1B, and the GSM8K dataset.

**Do we highly rely on good teacher generation?** As shown in Table 3, Llama3-70B achieves 93.18% accuracy on GSM8K and 56.85% on MMLU-Pro, while Llama3-8B achieves 80.97% on

GSM8K and 39.88% on MMLU-Pro. To evaluate the robustness of our method under varying teacher capabilities, we deliberately select teachers with different performance levels. The consistent improvements across both strong (70B) and weaker (8B) teacher models demonstrate that our approach is effective regardless of the teacher's skill level.

**Should we adopt adaptive evaluations for diverse tasks?** To assess the impact of different evaluation strategies in a verifiable task, we compare LLM-as-Judge and Majority Voting. As shown in Table 5, the Majority Voting consistently outperforms the LLM-as-Judge strategy across both models and both

Table 5: Class-wise precision of LLM evaluation strategies: LLM-as-Judge vs. Majority Voting.

| Model | Strategy | True | False |
|---|---|---|---|
| Llama3-70B | LLM-as-Judge | 95.8% | 67.8% |
| | Majority Voting | **97.5%** | **70.7%** |
| Llama3-8B | LLM-as-Judge | 86.6% | 48.0% |
| | Majority Voting | **95.7%** | **84.3%** |

classes. Notably, for LLaMA3-8B, the False-class precision improves significantly from 48.0% to 84.3%, demonstrating that voting mitigates the brittleness of smaller models. These results highlight the effectiveness of ensembling-based self-evaluation for more reliable pseudo-labeling. We provide a more detailed evaluation of majority voting in the Appendix E.4.

**How can we augment our reward model?** Table 6 presents an ablation study on different reward designs. Training an outcome reward model using classification loss on teacher responses (ORM-T) brings marginal improvement. However, training the ORM on student responses (ORM-S) leads to significantly better performance, showing the importance of distribution alignment. This difference can be attributed to the fact that reward models trained on the teacher's responses experience a greater distribution shift during policy learning,

Table 6: Ablation on Reward Design. CC represents 'Consistency Check'.

| Method | Acc (%) | vs. Ours w/o R |
|---|---|---|
| Ours w/o R | 63.08 | – |
| + ORM-T | 63.46 | +0.38 |
| + ORM-S | 64.90 | +1.82 |
| + ORM-S + CC | 65.06 | +1.98 |
| + Our RM | 65.70 | +2.62 |
| + Our RM + CC | 66.03 | +2.95 |

which adversely affects performance. Further gains are achieved by adding consistency checks and switching to our full reward model, which integrates preference supervision. The complete design achieves a +2.95% improvement over the no-reward baseline. We also investigate the design of the consistency check penalty in Appendix E.5.

**Other ablation studies and computational cost.** We include more comprehensive ablation studies in Appendix E: more detailed reward model performance evaluation, impact investigation of data ratio of teacher and student responses, comparison of using teacher and student responses for reward model training, robustness of hyperparameter $\lambda$ in Eq. 9, ablation of consistency check penalty in Eq 11 and Eq 10, as well as SFT result with more distillation data. We also include computational analysis in the Appendix F.

## 6 Conclusion

In summary, we propose a novel distillation framework that leverages both teacher-generated outputs and self-supervised reward signals to train a student model. By introducing reinforcement learning on top of SFT-based data distillation, this approach effectively sidesteps the biases of direct teacher evaluations and addresses the mismatch between the student model's inputs and the reward model in later training stages. Experimental results on GSM8K and MMLU-Pro demonstrate that this method not only outperforms purely SFT-based distillation strategies but also enables the student model to exceed the teacher's performance in certain metrics. Our work highlights the untapped potential of exploiting teacher LLMs' reward signals and offers a new, scalable paradigm for distilling large language models when reliable direct evaluation signals are absent.

**Limitation and Future Work.** Our approach relies on the teacher model's knowledge accumulated during its pretraining. We find that even with a 30% accuracy, we can learn a meaningful reward signal for the students. Future work will explore settings where the teacher is correct in only a small fraction of cases (e.g., 5%), aiming to maintain robust learning under sparse or noisy supervision and extend applicability to more challenging tasks.

**LLM Usage.** In addition to the usage of LLM described in the method, this paper uses LLM to help and polish the writing.

IMPACT STATEMENT

Our work seeks to contribute to the advancement of machine learning by enhancing the efficiency and scalability of knowledge distillation for large language models. By incorporating both generative outputs and self-supervised reward signals, our approach minimizes dependence on explicit teacher evaluations and human-labeled data. While this can improve accessibility and efficiency in model training, it also introduces challenges related to bias propagation and the reliability of self-supervised reward modeling. We recognize these concerns and encourage further research to ensure the robustness and fairness of such methods in real-world applications.

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

# A THEORETICAL ANALYSIS: IMPROVEMENT GUARANTEE OVER SUPERVISED DISTILLATION

We provide theoretical insights into our self-supervised reward distillation framework, showing that reinforcement learning (RL) guided by pseudo-rewards can provably improve student performance beyond supervised fine-tuning (SFT). Specifically, we consider a setup where the student policy $\pi_{\text{SFT}}$ is initialized via SFT on teacher-generated responses and subsequently updated via RL using a learned reward model trained on pseudo-labeled signals. Our analysis focuses on whether such updates lead to improvements with respect to the unknown true reward function $r^*$.

Assuming that the pseudo-reward function $r_p$ uniformly approximates the true reward within an error bound $\varepsilon$, we prove that the improvement in true expected return satisfies: $J_{r^*}(\pi_{\text{RL}}) - J_{r^*}(\pi_{\text{SFT}}) \geq \Delta_p - \frac{2\varepsilon}{1-\gamma}$, where $\Delta_p := J_{r_p}(\pi_{\text{RL}}) - J_{r_p}(\pi_{\text{SFT}})$ denotes the pseudo-reward improvement. This bound establishes that even without access to ground-truth reward supervision, the student policy can improve under the true objective as long as the pseudo-rewards are sufficiently accurate and the RL step is non-trivially beneficial. The detailed proof is provided below.

We provide a theoretical analysis to justify our core claim: that reinforcement learning (RL) with self-supervised pseudo-rewards can lead to a student policy that outperforms its supervised fine-tuning (SFT) initialization. Specifically, we show that under mild assumptions on pseudo-reward accuracy, a single RL update guided by these pseudo-rewards leads to an improvement in the true task return.

## A.1 PRELIMINARIES AND NOTATION

Let $\mathcal{D}$ denote the distribution over questions $q$, and let $\pi_{\text{SFT}}$ be the student policy obtained from supervised fine-tuning on teacher-generated answers. After an RL step using pseudo-rewards, we obtain an updated student policy $\pi_{\text{RL}}$. We use the following notation:

- $r^*(q, a) \in [0, 1]$: the unknown true reward for answer $a$ given input question $q$.
- $r_p(q, a)$: the pseudo-reward generated by our self-supervised mechanism.
- $Q_\pi^r(q, a)$: the expected return starting from answer $a$ on input $q$, under policy $\pi$ and reward function $r$.
- $J_r(\pi) = \mathbb{E}_{q \sim \mathcal{D}, a \sim \pi(\cdot|q)}[Q_\pi^r(q, a)]$: the expected return of policy $\pi$ under reward function $r$.
- $\gamma \in [0, 1)$: the discount factor for future rewards.

Our goal is to bound the improvement in true reward $J_{r^*}(\pi_{\text{RL}}) - J_{r^*}(\pi_{\text{SFT}})$ in terms of the pseudo-reward quality and policy change.

## A.2 PSEUDO-REWARD ACCURACY AND VALUE FUNCTION APPROXIMATION

We first show that if pseudo-rewards are close to the true rewards, the corresponding value functions will also be close.

**Lemma A.1 (Value-Function Approximation)** *Suppose that pseudo-rewards satisfy the uniform error bound:*

$$\sup_{q,a} |r_p(q, a) - r^*(q, a)| \leq \varepsilon.$$

*Then for any fixed policy $\pi$, the difference between the true and pseudo value functions is bounded:*

$$\|Q_\pi^{r^*} - Q_\pi^{r_p}\|_\infty \leq \frac{\varepsilon}{1-\gamma}.$$

**Proof A.2** *This follows from standard Bellman contraction arguments. Since the Bellman operators for $r^*$ and $r_p$ differ by at most $\varepsilon$ per step and are both $\gamma$-contractions, their fixed points (value functions) must differ by at most $\varepsilon/(1-\gamma)$.*

This lemma ensures that if our pseudo-rewards are accurate (i.e., small $\varepsilon$), the value estimates under $r_p$ are reliable approximations of those under the true reward $r^*$.

## A.3 Policy Improvement via Pseudo-Rewards

Let $\pi_{\mathrm{RL}}$ be the policy obtained after one policy improvement step from $\pi_{\mathrm{SFT}}$ using $r_p$. Define the *pseudo-advantage*:

$$\Delta_p := J_{r_p}(\pi_{\mathrm{RL}}) - J_{r_p}(\pi_{\mathrm{SFT}}).$$

This represents the improvement under the pseudo-reward due to the policy update. Standard RL methods (e.g., policy gradient, PPO) guarantee that $\Delta_p \geq 0$, and typically $\Delta_p > 0$ when learning progresses.

## A.4 Main Result: True Return Improvement Guarantee

We now present a detailed proof showing that reinforcement learning (RL) guided by pseudo-rewards leads to improvement in the true expected return, provided that the pseudo-rewards are sufficiently close to the true rewards.

**Theorem A.3 (True Return Improvement)** *Suppose the pseudo-reward function $r_p(q, a)$ satisfies the following uniform bound for all input-answer pairs:*

$$\sup_{q,a} |r_p(q, a) - r^*(q, a)| \leq \varepsilon.$$

*Let $\pi_{\mathrm{RL}}$ be the policy obtained by applying an RL update (e.g., policy gradient) to the SFT-initialized policy $\pi_{\mathrm{SFT}}$ using pseudo-reward $r_p$. Then the improvement in true expected return satisfies:*

$$J_{r^*}(\pi_{\mathrm{RL}}) - J_{r^*}(\pi_{\mathrm{SFT}}) \geq \Delta_p - \frac{2\varepsilon}{1 - \gamma},$$

*where $\Delta_p := J_{r_p}(\pi_{\mathrm{RL}}) - J_{r_p}(\pi_{\mathrm{SFT}}) \geq 0$ is the policy improvement under the pseudo-reward. Therefore, if $\Delta_p > \frac{2\varepsilon}{1-\gamma}$, then $\pi_{\mathrm{RL}}$ strictly improves over $\pi_{\mathrm{SFT}}$ under the true reward.*

**Proof A.4** *We begin by decomposing the difference in expected true return:*

$$\begin{aligned} J_{r^*}(\pi_{\mathrm{RL}}) - J_{r^*}(\pi_{\mathrm{SFT}}) &= \left[ J_{r_p}(\pi_{\mathrm{RL}}) - J_{r_p}(\pi_{\mathrm{SFT}}) \right] \\ &\quad + \left[ J_{r^*}(\pi_{\mathrm{RL}}) - J_{r_p}(\pi_{\mathrm{RL}}) \right] + \left[ J_{r_p}(\pi_{\mathrm{SFT}}) - J_{r^*}(\pi_{\mathrm{SFT}}) \right] \\ &= \Delta_p + \delta_1 + \delta_2. \end{aligned}$$

*We will bound the deviation terms $\delta_1$ and $\delta_2$. First, recall the definitions:*

$$J_r(\pi) = \mathbb{E}_{q \sim \mathcal{D}, a \sim \pi(\cdot|q)} \left[ Q_\pi^r(q, a) \right],$$

*and from Lemma A.1, we know that for any fixed policy $\pi$,*

$$\sup_{q,a} \left| Q_\pi^{r^*}(q, a) - Q_\pi^{r_p}(q, a) \right| \leq \frac{\varepsilon}{1 - \gamma}.$$

*Now, for any policy $\pi$, we can bound the difference in expected return:*

$$\begin{aligned} \left| J_{r^*}(\pi) - J_{r_p}(\pi) \right| &= \left| \mathbb{E}_{q \sim \mathcal{D}, a \sim \pi(\cdot|q)} \left[ Q_\pi^{r^*}(q, a) - Q_\pi^{r_p}(q, a) \right] \right| \\ &\leq \mathbb{E}_{q \sim \mathcal{D}, a \sim \pi(\cdot|q)} \left| Q_\pi^{r^*}(q, a) - Q_\pi^{r_p}(q, a) \right| \\ &\leq \sup_{q,a} \left| Q_\pi^{r^*}(q, a) - Q_\pi^{r_p}(q, a) \right| \leq \frac{\varepsilon}{1 - \gamma}. \end{aligned}$$

*Applying this bound to both policies $\pi_{\mathrm{RL}}$ and $\pi_{\mathrm{SFT}}$, we get:*

$$\left| J_{r^*}(\pi_{\mathrm{RL}}) - J_{r_p}(\pi_{\mathrm{RL}}) \right| \leq \frac{\varepsilon}{1 - \gamma}, \quad \left| J_{r_p}(\pi_{\mathrm{SFT}}) - J_{r^*}(\pi_{\mathrm{SFT}}) \right| \leq \frac{\varepsilon}{1 - \gamma}.$$

*Therefore, $\delta_1 := J_{r^*}(\pi_{\mathrm{RL}}) - J_{r_p}(\pi_{\mathrm{RL}}) \geq -\frac{\varepsilon}{1-\gamma}$, and similarly, $\delta_2 := J_{r_p}(\pi_{\mathrm{SFT}}) - J_{r^*}(\pi_{\mathrm{SFT}}) \geq -\frac{\varepsilon}{1-\gamma}$. Substituting back into the decomposition:*

$$J_{r^*}(\pi_{\mathrm{RL}}) - J_{r^*}(\pi_{\mathrm{SFT}}) \geq \Delta_p - \frac{2\varepsilon}{1 - \gamma}.$$

*This concludes the proof.*

This result shows that if the reward model trained from pseudo-labels is sufficiently accurate (i.e., small $\varepsilon$), and the RL update step improves the policy under pseudo-reward ($\Delta_p > 0$), then the student model will also improve under the true reward. In practice, we observe that even noisy pseudo-rewards—if consistent and sufficiently informative—can guide learning and enable the student to surpass the teacher's performance.

### A.5 EMPIRICAL JUSTIFICATION: REWARD MODEL ACCURACY

**Performance of Reward Model.** To validate the effectiveness of the reward model trained via our self-supervised mechanism, we evaluate its ability to distinguish between outputs of different quality. Specifically, we report binary classification accuracy on two tasks: (1) distinguishing between a positive teacher answer and a negative student answer (**Pos/Neg**), and (2) distinguishing between a positive teacher and a positive student answer (**Pos/Pos**) for the same question $q$.

As shown in Table 9, the reward model performs robustly across GSM8K and MMLU-Pro. It maintains strong accuracy even in the challenging **Pos/Pos** setting, demonstrating that it can identify subtle quality differences between high-performing models. These results empirically support the theoretical assumption of bounded pseudo-reward error ($\varepsilon$) and validate the feasibility of learning high-quality reward signals without external human labels.

## B PROMPTS IN EXPERIMENTS

### B.1 VERIFIABLE TASK

We provide prompts for collecting teacher's responses in Figure 3 (GSM8K) and Figure 4 (MMLU-Pro).

---

Q:
There are 15 trees in the grove. Grove workers will plant trees in the grove today. After they are done, there will be 21 trees. How many trees did the grove workers plant today?
A:
Let's break this down step by step!
Step 1: There are 15 trees originally.
Step 2: Then there were 21 trees after some more were planted.
Step 3: So there must have been 21 - 15 = 6.
The answer is 6.

Q:
If there are 3 cars in the parking lot and 2 more cars arrive, how many cars are in the parking lot?
A:
Let's break this down step by step!
Step 1: There are originally 3 cars.
Step 2: 2 more cars arrive, 3 + 2 = 5.
The answer is 5.

Q:
Leah had 32 chocolates and her sister had 42. If they ate 35, how many pieces do they have left in total?
A:
Let's break this down step by step!
Step 1: Originally, Leah had 32 chocolates.
Step 2: Her sister had 42.
Step 3: So in total they had 32 + 42 = 74.
Step 4: After eating 35, they had 74 - 35 = 39.
The answer is 39.

Q: {question}
Let's break this down step by step!

---

Figure 3: Prompt template for generating responses in the teacher LLMs over GSM8K dataset.

```
Question: {question of fewshot example 1}
Options: {options of fewshot example 1}
Answer: {answer of fewshot example 1}

Question: {question of fewshot example 2}
Options: {options of fewshot example 2}
Answer: {answer of fewshot example 2}

Question: {question}
Options: {options}
Answer:
Let's break this down step by step!
```

Figure 4: Prompt template for generating responses in the teacher LLMs over MMLU-Pro dataset. We use two shots (provided by the dataset) for few-shot learning.

### B.2 OPEN-ENDED TASK

We provide prompts of open-ended tasks, UltraFeedback for collecting teachers' responses, and evaluation in Figure 5 and Figure 6.

```
Question: {question}
Answer: {answer}
```

Figure 5: Prompt template for generating responses in the teacher LLMs over the UltraFeedback dataset.

## C IMPLEMENTATION DETAILS OF VERIFIABLE TASK

For our experiments, we implement the proposed method using a combination of different large language models (LLMs) as teachers, smaller models as students, and classifiers as reward models. **Teachers:** We employ Llama3-70B and Llama3-8B[1] (Dubey et al., 2024).This allows us to evaluate our method across teacher models with varying levels of knowledge. **Students:** We use Llama3-1Band Llama3-3B[2] (Dubey et al., 2024). These smaller models serve as students, learning and improving through knowledge distilled from the teacher models. **Reward Models:** The reward model is constructed from the student model by adding a single-head fully-connected layer on top of its final embedding layer, producing a scalar reward estimate. It takes both the question and the answer as inputs. **Data collection**, we prompt the teacher LLM five times for each query at temperatures 0, 0.1, 0.2, and 0.3 to collect high-confidence responses. For low-confidence responses, the teacher is prompted five times for each query, using the temperature set $\{0.1, 0.2, \cdots, 1\}$. For reward model learning, we infer the student model (after data distillation) 30 times per question. We set $\lambda$ in Eq. 9 to 0.5. Please refer to the Appendix C.2 for more details.

### C.1 DATA SPLIT

For GSM8K, we divided the original training dataset into training and validation sets, allocating 90% for training and 10% for validation.

For MMLU-Pro, we first allocate 15% of the data for testing. Then, we split the remaining data into training and validation sets using a 90% to 10% ratio.

---

[1]Llama-3-70B-Instruct, Llama3-80B-Instruct

[2]Llama3-3.2-1B-Instruct, Llama3-3.2-3B-Instruct

// System Prompt

Review the user's question and the corresponding response using the additive 5-point scoring system described below. Points are accumulated based on the satisfaction of each criterion:

1. Add 1 point if the response is relevant and provides some information related to the user's inquiry, even if it is incomplete or contains irrelevant content.
2. Add another point if the response directly addresses a large part of the user's question, even if some elements are missing or unclear.
3. Award a third point if the response answers the core aspects of the user's question in a generally useful way, even if it resembles blog posts or search results.
4. Grant a fourth point if the response is complete, coherent, and written from an AI Assistant's perspective, addressing the user's question directly and helpfully, with only minor room for improvement.
5. Bestow a fifth point for a response that is impeccably tailored to the user's question, demonstrating expert-level insight, clarity, and precision, without any extraneous content.

After reviewing the user's instruction and the response, first explain your reasoning for the score. Then, output the final total score.

Respond in the following format:

Evaluation evidence: <your brief explanation here, up to 100 words>
Score: <total score>

Assess from the perspective of an AI Assistant, using general web knowledge as needed. Evaluate strictly based on the five criteria above.

// User Prompt

<Question>: {question}
<Response>: {answer}

Figure 6: Prompt template for evaluating responses in the teacher LLMs over UltraFeedback dataset.

## C.2 HYPERPARAMETER

Our training pipeline consists of three stages: supervised fine-tuning (SFT) data distillation, reward model training, and proximal policy optimization (PPO). Each stage plays a critical role in progressively improving the student model.

**Dataset Specific Hyper-parameters**  We set the maximum generation length as 512 for GSM8K and 1024 for MMLU-Pro and 800 for UltraFeedback.

**Data distillation**  This Supervised Fine-Tuning (SFT) phase serves to initialize the student model prior to reinforcement learning. During SFT, we employ a learning rate of $5 \times 10^{-6}$ and a sequence length of 512 tokens. The batch size varies based on the specific teacher and student model configurations, as detailed in Table 7. Our dataset comprises majority-voted responses, ensuring a robust foundation for subsequent optimization. For the data distillation phase, we utilize 4 H100 GPUs and perform full parameter training. The training process spans 4 epochs, with checkpoints saved at intervals specified in the table. The optimal checkpoint is selected based on performance on the validation set. To accelerate training, we leverage DeepSpeed.

**Reward Model Training**  The reward model is trained to guide PPO-based fine-tuning. This stage uses a learning rate of $5 \times 10^{-5}$, a batch size of 48 for student Llama3-1B and a batch size of 16 for student Llama3-3B, and 4 training epochs. We apply early stop while the reward model performance

| Dataset | Teacher Model | Student Model | Batch Size | Save Steps |
|---------|---------------|---------------|------------|------------|
| GSM8K | Llama3-70B | Llama3-1B | 84 | 100 |
| | | Llama3-3B | 74 | 100 |
| | Llama3-8B | Llama3-1B | 84 | 100 |
| | | Llama3-3B | 70 | 100 |
| MMLU-Pro | Llama3-70B | Llama3-1B | 40 | 400 |
| | | Llama3-3B | 32 | 400 |
| | Llama3-8B | Llama3-1B | 40 | 100 |
| | | Llama3-3B | 32 | 100 |
| UltraFeedback | Llama3-70B | Llama3-1B | 48 | 100 |

Table 7: Batch size and checkpoint saving steps in data distillation phase.

stops increasing on the validationThe reward model is initialized from the student model after the warm-up. All reward models were trained on H100 GPUs.

**PPO Training**   The PPO stage refines the student model through reinforcement learning with the reward model. We use a learning rate of $1 \times 10^{-5}$, a KL penalty coefficient of 0.2, and a value function coefficient of 0.1. The total number of training episodes is set to $200,000$, ensuring sufficient interaction with the reward model for stable policy improvement. We apply early stop while the student model performance stops increasing on validation set (about 8k). We present more hyperparameters in Table 8.

| Dataset | Teacher Model | Student Model | Batch Size | Learning Rate | GPU_NUM |
|---------|---------------|---------------|------------|---------------|---------|
| GSM8K | Llama3-70B | Llama3-1B | 20 | $1 \times 10^{-5}$ | 2 |
| | | Llama3-3B | 4 | $5 \times 10^{-6}$ | 4 |
| | Llama3-8B | Llama3-1B | 20 | $1 \times 10^{-5}$ | 2 |
| | | Llama3-3B | 4 | $5 \times 10^{-6}$ | 4 |
| MMLU-Pro | Llama3-70B | Llama3-1B | 10 | $5 \times 10^{-6}$ | 4 |
| | | Llama3-3B | 2 | $1 \times 10^{-5}$ | 4 |
| | Llama3-8B | Llama3-1B | 10 | $1 \times 10^{-5}$ | 4 |
| | | Llama3-3B | 2 | $1 \times 10^{-5}$ | 4 |
| UltraFeedback | Llama3-70B | Llama3-1B | 16 | $2 \times 10^{-6}$ | 4 |

Table 8: Hyper-parameters in PPO.

**Ours w/o Data**   We apply a learning rate of $5 \times 10^{-5}$ and KL coefficient of 0.1 in this ablation version. Other hyper-parameters are the same to **PPO Training**.

# D   COMPARISON BETWEEN GKD AND OUR METHOD

The most significant difference lies in the assumption about access to internal logits: our method is designed to work without any access to teacher logits, whereas GKD Agarwal et al. (2024) explicitly relies on them, making it inapplicable in practical black-box teacher LLMs settings.

## D.1   DIFFERENCES OF THE REWARD SIGNAL BETWEEN GKD AND OURS

GKD discusses the potential of combining RLAIF-style training with their KL-based loss in Sec 3.2. However, the reward signals are provided by externally trained reward models built from human-annotated data (e.g., a textual entailment model for summarization), which goes beyond the distillation setting. In contrast, our method assumes we only have access to the teacher's outputs and the evaluation signals distilled from them, without relying on any additional human-labeled reward model.

Table 9: Accuracy of Reward Model to distinguish 1) positive teacher answer and negative student answer (Pos/Neg) and 2) positive teacher answer and positive student answer (Pos/Pos).

| $\mathcal{T}$ | $\mathcal{S}$ | GSM8K | | MMLU-Pro | |
|---|---|---|---|---|---|
| | | Pos/Neg | Pos/Pos | Pos/Neg | Pos/Pos |
| Llama3-70B | Llama3-1B | 0.7042 | 0.7223 | 0.6638 | 0.7752 |
| | Llama3-3B | 0.7081 | 0.7034 | 0.6670 | 0.9027 |
| Llama3-8B | Llama3-1B | 0.6573 | 0.8829 | 0.7417 | 0.8510 |
| | Llama3-3B | 0.6819 | 0.9382 | 0.7609 | 0.8798 |

### D.2 COMPLEXITY COMPARISON WITH GKD.

Our method is conceptually on a similar level of complexity as GKD:

- **Reward model training.** GKD relies on RLAIF and external reward models, which require collecting human preference data and training separate reward models. Our approach avoids this additional human supervision by constructing a reward model solely from the teacher and student outputs.

- **RL training.** The RL objective in GKD is non-differentiable, so GKD still requires an RL training phase comparable to ours. In this sense, their overall training pipeline is at least as complex as ours, even though it depends on extra human-labeled reward models, whereas ours operates purely in the teacher-distillation regime.

## E  ADDITIONAL EXPERIMENTAL RESULTS

### E.1  PERFORMANCE OF REWARD MODEL.

We report the performance of the reward model, distinguishing the positive and negative student responses (**Pos/Neg**) and the positive student and teacher responses (**Pos/Pos**), shown in Table 9.

### E.2  IMPACT OF DATA RATIO BETWEEN TEACHER'S AND STUDENT'S ANSWERS ON REWARD MODEL TRAINING.

We adjust the data ratio of teacher and student responses during reward model training. As illustrated in Figure 7, the data ratio has little effect on the reward model's ability to distinguish between good and bad responses but significantly influences its capacity to identify superior positive responses from both the teacher and the student.

### E.3  IMPACT OF HYPERPRAMETER $\lambda$ IN EQ. 9 ON REWARD MODEL'S PERFORMANCE.

We conduct an ablation study on the hyperparameter $\lambda$ to see the impact on the reward model's capacity of distinguishing the positive and negative answers. As shown in Figure 8, $\lambda = 0.5$ enables a more stable learning process.

### E.4  INVESTIGATE TEACHER'S SELF-EVALUATION ACROSS DATASETS

We visualize the teacher model's evaluation capability in Table 10 and Figure 9. Table 10 presents the proportion of questions for which the teacher fails to reliably generate a pseudo-final answer. The results indicate that while teacher models exhibit near-perfect evaluation on GSM8K, their performance drops significantly on MMLU-Pro. This issue is particularly evident with Llama3-8B, where nearly half of the questions fail to obtain a reliable pseudo-final answer. Furthermore, for questions with pseudo-final answers, we visualize the proportion of samples that the teacher model classifies as correct. As shown in Figure 9, the inaccuracy of the teacher's evaluation increases from Llama3-70B to Llama3-8B and from GSM8K to MMLU-Pro. In summary, evaluation capacity is weakest on MMLU-Pro, especially for Llama3-8B. Interestingly, as shown in Table 3, despite the most challenging setting—MMLU-Pro with Llama3-8B as the teacher—our method achieves

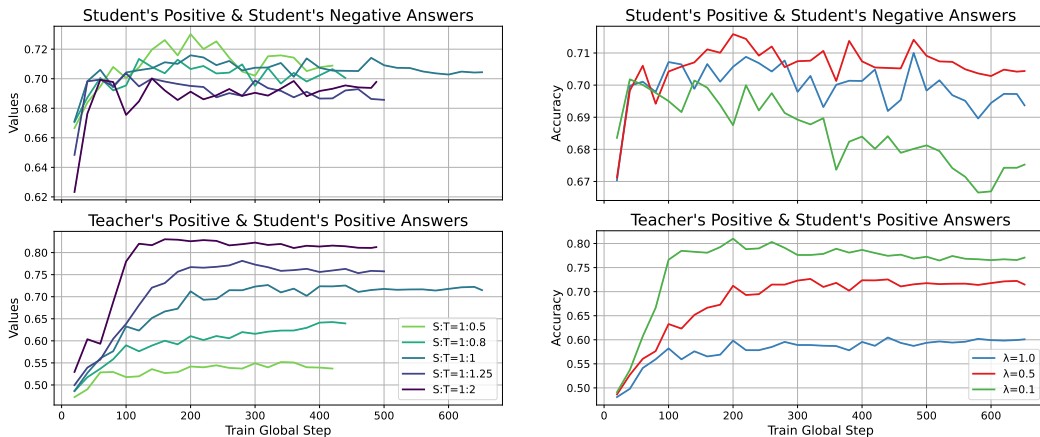

Figure 7: Reward model accuracy under varying ratios of teacher (T) and student (S) answers during training. Darker color indicate a higher proportion of teacher answers.

Figure 8: Accuracy of the reward model under different values of $\lambda$ in Eq. 9.

Table 10: Proportion of questions where the teacher fails to produce a pseudo label via majority voting, due to no candidate answer surpassing the predefined voting threshold.

| Dataset | LLaMA3-70B | LLaMA3-8B |
|---------|------------|-----------|
| GSM8K | 0.03 | 0.10 |
| MMLU_PRO | 0.23 | 0.48 |

the largest performance improvement, highlighting its robustness in scenarios with weaker teacher supervision. The probable reason behind this is that the student gains the least knowledge from data distillation, therefore leaving much room for improvement.

### E.5    ABLATION ON DESIGN OF CONSISTENCY CHECK PENALITY IN EQ 10 AND EQ. 11

We conduct an ablation study on methods to correct the reward model's predictions. In our full method, we apply $\min(r, 0)$ when the extracted answer $y$ differs from the pseudo label $y^*$ (**Ours**). Additionally, we evaluate ablation variants that apply $r - 1$ when $y \neq y^*$ (**Minus**), and apply no consistency check (**None**).

### E.6    PERFORMANCE OF DIRECT PREFERENCE OPTIMIZATION

We also provide the experimental results of Direct Preference Optimization (DPO), while using LLaMA3-8B as the teacher and LLaMA3-1B as the student.

As shown in Table 11, DPO underperforms compared to our method. One possible explanation is that **DPO relies heavily on high-quality preference data**. In contrast, except for reward modelling, our framework allows for the **straightforward incorporation of additional constraints** into the reward signal during training, which can further enhance the student model's performance. Moreover, although we can naturally generate preference pairs through self-evaluation, standard DPO requires explicit preference annotations, which can incur substantial computational and annotation costs.

### E.7    CAN MORE SFT DATA FURTHER INCREASE THE STUDENT'S PERFORMANCE?

Figure 11 shows that increasing the amount of SFT data generally improves student performance during data distillation, but the gain saturates beyond the scale used in our method.

## F    COMPUTATIONAL COST ANALYSIS

We provide a concrete computational cost analysis on GSM8K below:

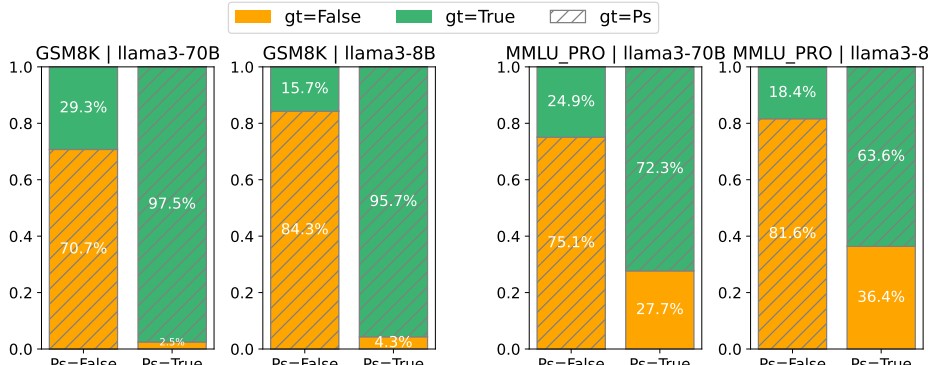

Figure 9: Illustration of the teacher's self-evaluation accuracy. In each subfigure, answers identified as incorrect by the teacher are shown on the left (Ps = False), and those identified as correct are on the right (Ps = True). The ground-truth correctness of each answer is indicated by color: green for True and orange for False. Shaded regions highlight the cases where the teacher's self-evaluation aligns with the ground-truth.

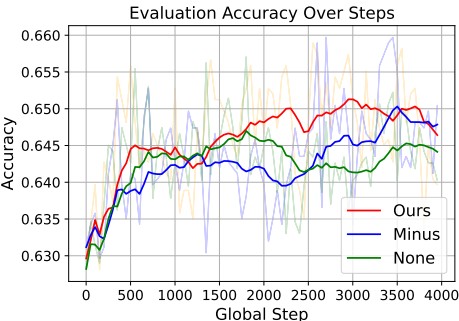

Figure 10: Comparison of consistency strategies with the pseudo final answer.

- Teacher generation + warm-up SFT. This does not incur additional cost beyond standard SFT, since it is a shared phase with conventional supervised distillation pipelines.
- Student generation (for reward model training):
  - Llama-1B student: inference once over the selected data on 1 H100 takes about 6 min. For 30 answers, 6 min × 30 = 3 hours.
  - Llama-3B student: inference once on 1 H100 takes about 0.25 hours. For 30 answers, 0.25 hours × 30 = 7.5 hours.

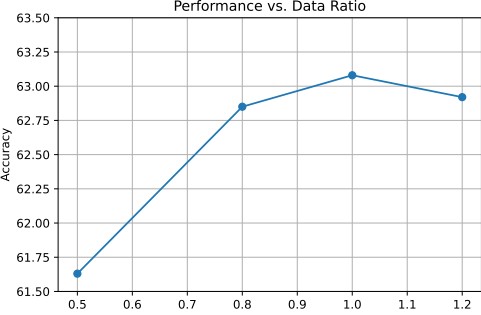

Figure 11: Performance of SFT data distillation under different training data scales. Data amounts are normalized relative to the dataset size used in our method.

Table 11: Comparison of Optimization Objectives

|            | SFT   | DPO   | PPO (Ours) |
|------------|-------|-------|------------|
| Accuracy (%) | 63.08 | 63.46 | 66.03 |

- Reward model training (distilling reward signals from the teacher's responses):
  - Llama-1B reward model: on 1 H100, 4 epochs cost 1 hour.
  - Llama-3B reward model: on 4 H100, 4 epochs cost 4 hours.
- PPO training:
  - Llama-1B student: on 4 H100, about 8 hours for $5k$ steps.
  - Llama-3B student: on 4 H100, about 32 hours for $8k$ steps.

## G  EXAMPLE OF $\mathcal{P}(q)$

$D_R$ contains questions paired with three types of answers:

- **Question:** `"1/3 of the townspeople have received the full COVID vaccine..., what percent of the town is immune in some way?"`
- **Positive student answer** $a^{S^+}$ and label $y^{S^+}$:
  - $a^{S^+}$: `Step 1:  1/3 of the townspeople ...  Step 5:  33.33 + 33.33 - 16.67 = 50.`
    `The answer is 50%.`
  - $y^{S^+} = 50$
- **Negative student answer** $a^{S^-}$ and label $y^{S^-}$:
  - $a^{S^-}$: `Step 1:  1/3 of the townspeople ...  Step 5:  2/3 * 100 = (2*100)/3 = 200/3 = 66.67%.`
    `The answer is 66.67%.`
  - $y^{S^-} = 66.67$
- **Positive teacher answer** $a^{T^+}$ and label $y^{T^+}$:
  - $a^{T^+}$: `Step 1:  1/3 of the townspeople have received the full COVID vaccine...  Step 4:  percentage that's double-counted:  33.33 + 33.33 - 16.67 = 50.`
    `The answer is 50%.`
  - $y^{T^+} = 50$

