# OpenReview forum: "Distill Not Only Data but Also Rewards: Can Smaller Language Models Surpass Larger Ones?"
_ICLR.cc/2026/Conference — Submitted to ICLR 2026_

### Official Review · Reviewer_5AyG · 2025-10-24

**Soundness:** 3
**Presentation:** 2
**Contribution:** 2
**Rating:** 4
**Confidence:** 4

**Summary:**

This paper proposes a novel KD with RL framework where a student model learns not only from the teacher’s responses but also from the teacher’s evaluation signals, i.e., rewards. The core idea is to train a reward model that aligns with the teacher’s preferences, allowing student outputs to be optimized via PPO against this learned reward signals. The authors also propose separate reward modeling strategies for verifiable and open-ended tasks. Experiments on GSM and MMLU benchmarks show improvements over standard distillation baselines.

**Strengths:**

- The central idea of jointly distilling both response and reward signals from a teacher LLM is novel and appears effective.
- Separate reward modeling for verifiable tasks and open-ended tasks is a well-motivated design, highlighting that evaluation criteria differ depending on task types.

**Weaknesses:**

While the motivation and method are promising, the paper lacks deeper analysis to justify whether this RL-based approach is really superior over existing KD methods.

- A core assumption in this paper is that LLMs are unreliable evaluators, which motivates training an explicit reward model. However, training the reward model involves constructing the reward dataset, which itself is built from the teacher LLM judgments of teacher and student responses. If these evaluations are unreliable as claimed, the reward model may inherit their noise and inconsistencies. And if the teacher LLM is good enough to correctly evaluate, there is no need to introduce another reward model. Thus, I’m curious if this reward model has been properly constructed.
- The experimental comparison is limited. One of the primary baselines (**Ours w/o R**) only reflects basic response-level KD using teacher outputs. However, existing KD methods like ImitKD (Lin et al., 2020), GKD (Agarwal et al., 2024), and DistiLLM (Ko et al., 2024) also incorporate student outputs and often use adaptive incorporation with teacher responses. Since the proposed PPO (Eq. 13) uses dataset $\mathcal{D}_{\mathcal{R}}$ which includes responses from both teacher and student, for fair comparison, the paper should include stronger, state-of-the-art KD baselines that similarly exploit student trajectories.

**Questions:**

- It is unclear how the method performs when the teacher is incapable of providing high-quality responses (i.e., none of the responses reach the target reward $\ell^*$). In such cases, the teacher-student preference dataset $\mathcal{D}_{\mathcal{R}}$ may be skewed toward simpler tasks, limiting the effectiveness of reward model training. Can the method still construct a meaningful reward dataset for those queries?
- Is the reward model robust to noisy or inconsistent teacher scores? How is this addressed during training?

**Minor typos**

- $D_{\text{TeachCan}}$ in Eq. 3
- *Resul* in Sec. 5.3 header

---

> ### Author Response · Authors · 2025-11-18
>
> We thank the reviewer for the time and effort in reviewing our paper. We also appreciate the recognition of the novelty in jointly distilling responses and reward signals with task-specific designs.
>
> ---
>
> ## Response to Weakness 1
> > While the motivation and method are promising, the paper lacks deeper analysis to justify whether this RL-based approach is really superior over existing KD methods.
>
> We first clarify our scope: **we focus on distillation in the setting where the teacher’s internal token-level logits are not accessible**, which is increasingly common in the current LLM ecosystem—teachers are often exposed only as black-box APIs, or online querying of a large teacher model is prohibitively expensive.
>
> **This setting directly determines our choice of only baseline: supervised fine-tuning,** since other KD methods that rely on the teacher’s internal logits are not applicable here.
>
> Within this constrained but practical setup, our experimental results and accompanying analysis provide clear evidence that the proposed RL-based distillation consistently outperforms the SFT baseline across our evaluated tasks.
>
> ---
> ## Response to Weakness 2 and Question 2
> > A core assumption in this paper is that LLMs are unreliable evaluators, which motivates training an explicit reward model. However, training the reward model involves constructing the reward dataset, which itself is built from the teacher LLM judgments of teacher and student responses. If these evaluations are unreliable as claimed, the reward model may inherit their noise and inconsistencies. And if the teacher LLM is good enough to correctly evaluate, there is no need to introduce another reward model. Thus, I’m curious if this reward model has been properly constructed.
> >
> > Question 1: Is the reward model robust to noisy or inconsistent teacher scores? How is this addressed during training?
>
>
> Thank you for raising this point. **Our assumption is not that LLMs are unusably unreliable evaluators, but that their raw evaluations contain noise, and naïvely using them online during RL is unstable.** Our method addresses exactly this issue by denoising and stabilizing these evaluations into a consistent reward function.
>
> First, Table 1 shows that even imperfect teacher evaluations lead to clear, consistent gains once distilled through our reward model. This empirically confirms that noisy but informative labels can still produce meaningful reward supervision.
>
> Second, we explicitly reduce noise before and during reward modeling:
> - **Adaptive evaluation:** For verifiable tasks, majority voting drastically improves precision; for open-ended tasks, LLM-as-Judge is used. Table 5 shows this significantly boosts reliability.
> - **Filtering unreliable cases:** We discard low-confidence teacher evaluations and construct balanced datasets, removing the worst noise.
> - **Consistency correction:** During RL, if the reward model disagrees with the pseudo-final answer in verifiable tasks, we correct its output (Eq. 11), further stabilizing the signal.
>
> These steps ensure that the reward model does not simply inherit raw teacher noise, but instead learns a cleaner, more consistent evaluative signal.
>
> Finally, **the reward model’s quality is validated directly by the results**: across multiple datasets, teacher scales, and student sizes, the distilled reward model consistently improves the student, demonstrating that it is properly constructed and provides reliable training feedback.
>
> ---
>
> ## Response to Weakness 3:
> > The experimental comparison is limited. One of the primary baselines (Ours w/o R) only reflects basic response-level KD using teacher outputs. However, existing KD methods like ImitKD (Lin et al., 2020), GKD (Agarwal et al., 2024), and DistiLLM (Ko et al., 2024) also incorporate student outputs and often use adaptive incorporation with teacher responses. Since the proposed PPO (Eq. 13) uses dataset which includes responses from both teacher and student, for fair comparison, the paper should include stronger, state-of-the-art KD baselines that similarly exploit student trajectories.
>
> As mentioned in our response to Weakness 1, we focus on the setting where the **teacher LLM’s internal token-level logits are inaccessible.** This is increasingly common in the current LLM era: the teacher is often a very large model (up to 70B parameters in our experiments) whose online inference is extremely costly, or even exposed only as a black-box API.
>
> Under this constraint, **methods such as ImitKD, GKD, and DistiLLM are not applicable**, because they all explicitly rely on access to the teacher’s logits/probabilities to construct their KD objectives or adaptive mixing strategies. For this reason, we cannot fairly compare to these methods in our setting, and instead use response-level SFT as baselines that are compatible with a black-box teacher.

---

> ### Author Response · Authors · 2025-11-18
>
> ## Response to Question 1:
> > It is unclear how the method performs when the teacher is incapable of providing high-quality responses (i.e., none of the responses reach the target reward ). In such cases, the teacher-student preference dataset may be skewed toward simpler tasks, limiting the effectiveness of reward model training. Can the method still construct a meaningful reward dataset for those queries?
>
>
> We clarify that our reward model dataset is only constructed from queries that the teacher is able to answer (as illustrated in Sec 4.2), and all preference pairs are formed within the same query (teacher vs. student for that query). Thus, the preference data are always matched in difficulty and are not skewed toward simpler tasks.
>
> Regarding the teacher’s capability:
> - Partial capability (our setting): When a subset of queries in the dataset can reach the target reward, this is the typical and also our experimental setting. We select those solvable queries for reward modeling and student training.
> - No capability: If none of the queries can reach the target reward, this indicates that the task is already too difficult for the teacher, leaving no meaningful knowledge to transfer. In such a regime, the student would not learn from the teacher using any teacher–student distillation method, including ours.
>
> ---
>
> ## Minor typos
> Thanks for pointing those out. We've revised in the attached version.
>
>
> ---
> We hope that our clarifications and revisions adequately address the concerns raised, and we kindly request the reviewer to reconsider the overall rating; we are sincerely grateful once again for the assessment of our work.

---

### Official Review · Reviewer_5QHe · 2025-10-31

**Soundness:** 2
**Presentation:** 3
**Contribution:** 2
**Rating:** 4
**Confidence:** 3

**Summary:**

This paper proposes a unified distillation framework for LLMs, arguing that traditional SFT (data distillation) is limited because it propagates teacher errors and cannot surpass imitation. The authors introduce reward distillation, which extracts evaluation signals from the teacher without human labels. This is done using majority voting for verifiable tasks and LLM-as-Judge for open-ended ones. A student model is first warmed up with SFT, then refined using reinforcement learning (RL) guided by a learned reward model. This combined approach consistently outperforms SFT, allowing student models to in some cases surpass their teachers.

**Strengths:**

1. Novel Annotation-Free Framework: The paper proposes a novel reward distillation framework. This method cleverly extracts evaluation signals directly from the teacher (via majority voting or LLM-as-Judge) without needing expensive human annotations.
2. Clear and Easy to Follow: The paper is well-structured and clearly written, making its complex methodology understandable.

**Weaknesses:**

A primary weakness is the claim that the student can surpass the teacher. This is likely because the student model is trained on the GSM8K and MMLU-Pro data. Even though the teacher generated these answers and majority voting was used to enhance the data quality, the teacher's baseline performance is being compared against a student's fine-tuned performance. A more reasonable comparison would require fine-tuning the teacher model on the exact same high-quality, distilled dataset and then comparing its performance to the student's.

**Questions:**

N/A

---

> ### Author Response · Authors · 2025-11-18
>
> We thank the reviewer for the thoughtful feedback. We are encouraged that the reviewer finds our annotation-free reward distillation framework novel and appreciates the clarity and structure of the paper.
>
> ---
>
> ## Response to the Weaknesses:
> > A primary weakness is the claim that the student can surpass the teacher. This is likely because the student model is trained on the GSM8K and MMLU-Pro data. Even though the teacher generated these answers and majority voting was used to enhance the data quality, the teacher's baseline performance is being compared against a student's fine-tuned performance. A more reasonable comparison would require fine-tuning the teacher model on the exact same high-quality, distilled dataset and then comparing its performance to the student's.
>
> This concern may arise from the **misunderstanding of our problem**. We focus on the distillation setting, where a fixed, stronger teacher is used to train a smaller student. In this scenario, **only the student is trainable, while the teacher (often a larger model) is kept frozen and cannot be further fine-tuned.** Therefore, fine-tuning the teacher on the same distilled dataset and comparing it with the student falls outside our problem setting, which focuses on compressing a fixed teacher into a smaller, deployable student. We clarify this assumption more explicitly in the revised version.
>
> ---
>
> We hope that our responses sufficiently address your concern, and we would be grateful if you could kindly reconsider the overall rating; we sincerely thank you again for your feedback.

---

### Official Review · Reviewer_X6QW · 2025-11-02

**Soundness:** 2
**Presentation:** 3
**Contribution:** 2
**Rating:** 2
**Confidence:** 4

**Summary:**

- This paper introduces a novel distillation method to transfer the knowledge from the responses, but also the evaluative capabilities from the rewards.
- The proposed reward distillation method uses a self-supervised process within the teacher-student framework. First, it uses the teacher LLM to generate a variety of responses. Then, it employs an adaptive evaluation strategy to create pseudo labels: majority voting or LLM-as-a-judge. A reward model is then trained on a mix of teacher and student-generated responses to learn these reward signals. Finally, the student model, after an initial warm-up with supervised fine-tuning, is further refined using reinforcement learning guided by this distilled reward model.

**Strengths:**

- The paper is well-written and easy to follow.
- The core motivation of this paper is to mitigate the distribution shift between the teacher and student, which is a critical challenge and a good research question in knowledge distillation.

**Weaknesses:**

- For On-Policy RL (PPO), since the rollouts are sampled from the student policy model, there would not exist any distribution shift between the policy and the experience. Therefore, the proposed method does not solve a real question for On-Policy RL training.
- For the verifiable task, this work should compare the proposed Reward Distillation with the standard RLVR method. Why is Reward Distillation better than RLVR?
- For the non-verifiable task, this work should compare the proposed Reward Distillation with the RLHF with GenRM (LLM-as-a-Judge). Why is Reward Distillation better than RLHF with GenRM?
- The paper positions itself primarily against traditional distillation but does not sufficiently discuss or compare against the closely related paradigm of self-rewarding or self-improving LLMs[1][2]. These methods also involve a model generating its own training signals or rewards.
- For the reward signal, using the teacher model's signal as a reward has been proposed by On-Policy Distill[3][4], which can provide dense reward signals in the sequence dimension. However, the proposed Reward Distillation only provides an outcome reward signal and should construct data to train an additional reward model.

[1] Self-Rewarding Language Models

[2] Meta-Rewarding Language Models: Self-Improving Alignment with LLM-as-a-Meta-Judge

[3] On-Policy Distillation of Language Models: Learning from Self-Generated Mistakes

[4] MiniLLM: Knowledge Distillation of Large Language Models

**Questions:**

Same as the weaknesses.

---

> ### Author Response · Authors · 2025-11-18
>
> Thank you for raising concerns and detailed questions. We appreciate your recognition that our method is clearly presented and easy to follow. Below we address your concerns in turn.
>
> ---
>
> ## Response to Weakness 1:
> > For On-Policy RL (PPO), since the rollouts are sampled from the student policy model, there would not exist any distribution shift between the policy and the experience. Therefore, the proposed method does not solve a real question for On-Policy RL training.
>
> Our policy learning indeed uses an on-policy algorithm, PPO, but the reward model is pre-trained offline (as in RLHF/RLAIF).
> **This creates a different source of distribution shift:**
> - If the reward model is trained only on the teacher’s responses, then during PPO, the online rollouts come from the student, while the reward model has only been exposed to the teacher distribution. **This mismatch induces a distribution shift between reward training data and on-policy trajectories.**
>
> To address this, **we mainly train the reward model on the student’s generations**, so that its training distribution matches the on-policy data. To still exploit the stronger teacher, we further **augment the reward loss with a preference loss that prefers the teacher’s responses over the student’s**, achieving reduced distribution shift with effective teacher guidance.
>
> ---
>
> ## Response to Weakness 2:
> > For the verifiable task, this work should compare the proposed Reward Distillation with the standard RLVR method. Why is Reward Distillation better than RLVR?
> For the non-verifiable task, this work should compare the proposed Reward Distillation with the RLHF with GenRM (LLM-as-a-Judge). Why is Reward Distillation better than RLHF with GenRM?
>
> Thanks for the reviewer's question. Our method is to distill both the teacher’s generation and its reward signal into a smaller student, rather than distilling the reward signal.
>
>
> **Our setting is different from RLVR and RLHF with GenRM, so they are not directly comparable:**
> - We explicitly study a distillation problem: distilling a large teacher into a smaller student without access to the teacher’s token-level logits (black-box teacher).
> - Both RLVR and RLHF with GenRM instead assume direct RL on a single model with an external verifiable reward (RLVR) or a separately trained GenRM / human data, rather than using a teacher–student setup.
>
> In other words, RLVR and RLHF+GenRM target alignment with external rewards, while our work targets teacher–student distillation under restricted teacher access, which is a different problem.
>
> ---
>
> ## Response to Weakness 3:
> > The paper positions itself primarily against traditional distillation but does not sufficiently discuss or compare against the closely related paradigm of self-rewarding or self-improving LLMs[1][2]. These methods also involve a model generating its own training signals or rewards.
>
>
> Thank you for the question. We agree that self-rewarding and self-improving LLMs are important developments, and we have added a brief discussion of [1,2] in the Related Work section. However, **these methods are not comparable to our approach because they operate under a fundamentally different problem setting.**
>
> Our work studies teacher–student distillation, where **the student is significantly weaker and relies on a stronger external teacher to provide both data and evaluative signals.** The entire goal of our framework is to distill the teacher’s generation and reward signals into a smaller student model.
>
> Self-rewarding/self-improving methods, in contrast, **assume a single sufficiently strong model that can generate both its own outputs and its own training signals without any external teacher.** These methods do not address the core challenge we study—how to transfer evaluative ability from a large teacher to a smaller student in a logit-free setting—and therefore lie outside the scope of distillation.
>
> For these reasons, a direct comparison would not be meaningful, and these approaches are not alternatives to our setting. We have clarified this distinction in the revised manuscript.

---

> ### Author Response · Authors · 2025-11-18
>
> ## Response to Weakness 4:
> > For the reward signal, using the teacher model's signal as a reward has been proposed by On-Policy Distill[3][4], which can provide dense reward signals in the sequence dimension. However, the proposed Reward Distillation only provides an outcome reward signal and should construct data to train an additional reward model.
>
> Thanks for the question. Our setting differs from On-Policy Distill [3,4], leading to a different design.
>
> **Difference in access to internal logits.** On-Policy Distill uses the teacher’s token-level signals as dense supervision, which requires access to the teacher’s internal logits/probabilities. In our setting, the teacher is a black-box large LLM, so such dense token-level signals are not available.
>
> **Difference in learning objective:**
> - In On-Policy Distill, token-level rewards focus on blindly imitating the teacher’s behavior, which can be harmful when the teacher produces low-quality generations.
> - In contrast, we explicitly distill task-relevant outcome rewards from the teacher’s evaluations and then optimize the student with RL on this reward, allowing the student to learn from both good and bad generations rather than performing per-token mimicry.
>
> Therefore, although On-Policy Distill indeed produces denser rewards, they are not directly optimized for downstream task performance. By contrast, our method directly optimizes task performance and avoids the huge cost of online inference of the large teacher model.
>
> ---
>
> We hope that the clarifications and additional analysis provided in our rebuttal help resolve the concerns and questions you raised, and we respectfully request that you reconsider the rating. We sincerely thank you again for your careful review!

---

### Official Review · Reviewer_Ko1H · 2025-11-05

**Soundness:** 2
**Presentation:** 2
**Contribution:** 1
**Rating:** 2
**Confidence:** 4

**Summary:**

This paper proposes a multi-stage distillation pipeline as an alternative to standard supervised fine-tuning (SFT). The core idea is to distill not only the teacher's data (responses) but also its evaluative capacity (rewards). The method involves (1) an SFT warm-up, (2) training a separate, small reward model (RM) using teacher-generated labels (via majority vote or LLM-as-Judge), and (3) refining the student using Proximal Policy Optimization (PPO) guided by this RM.

**Strengths:**

The paper's primary strength is its well-motivated strategy of leveraging both the teacher's generative and evaluative capabilities. Empirical results demonstrate that this pipeline outperforms SFT. Warm-up training can further improve results (although same finding has been pointed out by GKD, SKD paper).

GKD: https://arxiv.org/pdf/2306.13649
SKD: https://arxiv.org/abs/2410.11325

**Weaknesses:**

W1: Limited Novelty and Missing Baselines: This is the most significant weakness. The core idea of using reinforcement learning (via policy gradient) and reward signals for distillation is not new (e.g., GKD, MiniLLM, direct preference knowledge distillation). The paper's novelty is a specific pipeline, yet it fails to compare against these established RL-based distillation baselines. The comparison is limited to SFT, which is a weak baseline making it impossible to assess if this complex SFT+RM+PPO pipeline is any better than simpler, existing RL-KD methods or simply KL loss + reward objective.

W2: The contribution of the reward signal generation is overstated. The components are well-known: "majority voting" is a direct application of Self-Consistency, and "LLM-as-Judge" is a standard technique. The paper's contribution is a synthesis of these existing methods, not a novel mechanism.

W3: The paper's pipeline (SFT -> RM training -> PPO) is significantly complex, but its design choices are poorly motivated. PPO vs. GKD: It is unclear why this complex, multi-stage pipeline is necessary over simpler, end-to-end objectives like GKD, which also incorporate reward signals. The paper provides no justification.

W4: Separate RM: The paper trains a separate, small RM but never justifies why this is necessary over using the teacher model as the reward source. This is a critical unstated detail (likely for computational cost), which also highlights the next weakness. If RM is expensive, why don't authors consider direct preference KD?

W5: Unalyzed Computational Cost: The paper ignores the massive computational cost of its pipeline. The reward generation requires 10 teacher samples + 30 student samples and evaluations per query. This cost is not analyzed, and it's plausible that a baseline SFT model trained with the same compute budget (i.e., 40x more data) would perform just as well.

GKD: https://arxiv.org/pdf/2306.13649
miniLLM: https://arxiv.org/abs/2306.08543
Direct preference KD: https://arxiv.org/abs/2406.19774

**Questions:**

Why training a separate RM?

---

> ### Author Response · Authors · 2025-11-18
>
> Thank you for the detailed and critical review. We appreciate your recognition of our well-motivated method and the empirical results. Below, we address your concerns in turn.
>
> ## Response to Weaknesses 1:
> > W1: Limited Novelty and Missing Baselines: This is the most significant weakness. The core idea of using reinforcement learning (via policy gradient) and reward signals for distillation is not new (e.g., GKD, MiniLLM, direct preference knowledge distillation). The paper's novelty is a specific pipeline, yet it fails to compare against these established RL-based distillation baselines. The comparison is limited to SFT, which is a weak baseline, making it impossible to assess if this complex SFT+RM+PPO pipeline is any better than simpler, existing RL-KD methods or simply KL loss + reward objective.
>
>
> We thank the reviewer for raising this important point. We believe this concern primarily stems from **a misunderstanding of our problem setting and the scope explicitly stated in the paper.**
>
> ### Our Scope: distillation without access to logits
> **Our setting fundamentally differs from logit-based KD methods such as GKD, MiniLLM, and direct preference KD.** Our work explicitly targets the increasingly common practical scenario where the teacher is a black-box LLM exposed only through text APIs, and token-level logits are not available. This constraint is emphasized in both the Introduction (Lines 51–53) and Related Work (Lines 101–104), and we highlight it more clearly in the revised version.
>
> Because of this logit-free setting, methods like GKD, MiniLLM, DPKD, etc. are not applicable, as they require: direct access to teacher logits, KL-style divergences between teacher and student, and often the same tokenizer between teacher and student. These assumptions do not hold in modern LLM ecosystems, where teachers are often commercial systems (GPT-like APIs) or prohibitively large models (8B–70B in our experiments). Therefore, including these methods as baselines would be misleading and incompatible with our setting.
>
> ### Logit-based KD is not RL-based
>
> In addition, the reviewer’s suggested baselines (GKD, MiniLLM, direct preference KD, etc.) are not RL-based distillation methods in the sense relevant to our framework. Despite occasional references to RL, their training objective is still a supervised KL-style divergence toward the teacher distribution, meaning the student is optimized to mimic the teacher’s outputs rather than to optimize any task-level reward. Our approach fundamentally differs: the student is trained with an explicit, task-relevant reward signal distilled from the teacher’s evaluations, enabling optimization of answer quality rather than distribution matching. Because these methods target imitation under logit access, they address different problems and are not appropriate baselines for our scenario.
>
> ### More specific comparison of GKD and ours
> #### Reward signal.
> GKD discusses the potential of combining RLAIF-style training with their KL-based loss in Sec 3.2. However, the reward signals are provided by externally trained reward models built from human-annotated data (e.g., a textual entailment model for summarization), which goes beyond the distillation setting. In contrast, **our method assumes we only have access to the teacher’s outputs and the evaluation signals distilled from them**, without relying on any additional human-labeled reward model.
>
> #### Complexity
>
> **Regarding complexity, our method is not more complex than the reviewer’s suggested GKD with reward signals, named GKD + RLAIF/RLHF in the GKD paper.** GKD + RLAIF/RLHF requires training external reward models using human preference data and then performing RL on top of those models, since its RL objective remains non-differentiable. In contrast, our framework stays entirely within the distillation setting: all reward signals are derived automatically from teacher and student textual outputs, and no additional human-labeled reward model is needed. The resulting RL stage is comparable in structure to GKD’s, but the overall pipeline is actually simpler because it eliminates the extra human-supervised components. Thus, our approach does not introduce additional complexity beyond what GKD already requires.
>
>
>
> ### Our Revision
>
> We would like to note that **the original submission already explained this scope and clarified that, under our logit-free setting, SFT is the only applicable baseline** (Introduction Lines 51–53 and Related Work Lines 101–104). To address the reviewer’s concern more clearly, we have further revised the abstract, introduction, and appendix to explicitly articulate these distinctions, highlight why logit-based KD methods are not comparable, and more clearly emphasize the intended scope of our work. All corresponding updates are marked in blue in the revised manuscript.

---

> ### Author Response · Authors · 2025-11-18
>
> ## Response to Weaknesses 3:
> > W3: The paper's pipeline (SFT -> RM training -> PPO) is significantly complex, but its design choices are poorly motivated. PPO vs. GKD: It is unclear why this complex, multi-stage pipeline is necessary over simpler, end-to-end objectives like GKD, which also incorporate reward signals. The paper provides no justification.
>
>
>
> Thank you for the question. **Our pipeline follows directly from the central goal of the paper: distill not only data but also reward.** Achieving this goal requires separating generation learning and reward learning, which **GKD-style end-to-end objectives cannot support.**
>
> 1) First, GKD is not applicable in our setting: it depends on teacher logits and KL-based imitation losses, which are impossible to compute in our logit-free, black-box teacher scenario.
> 2) Second, GKD does not optimize a real task reward. Its objective still measures how well the student mimics teacher outputs, so it may propagate teacher errors, contrary to our aim of teaching the student “what is preferred,” not just “what the teacher said.”
> 3) Third, when GKD introduces a reward, it relies on externally trained, human-labeled reward models and still requires RL to optimize a non-differentiable signal, violating the distillation assumption that only a teacher model should be available.
>
> Thus, our SFT → RM → PPO pipeline is not excessive; it is **the minimal structure** needed to (1) warm up the student, (2) distill the teacher’s self-evaluation into an explicit task-aware reward model, and (3) optimize the student with respect to this reward, precisely enabling us to distill both responses and evaluative ability rather than mere imitation.
>
> We further clarified this motivation in the revised manuscript.
>
>
>
> ---
>
> ## Response to Weaknesses 4 and Question:
> > W4: Separate RM: The paper trains a separate, small RM but never justifies why this is necessary over using the teacher model as the reward source. This is a critical unstated detail (likely for computational cost), which also highlights the next weakness. If RM is expensive, why don't authors consider direct preference KD?
> >
> > Question: Why training a separate RM?
>
> Thank you for the question. The need for a separate reward model follows directly from our core motivation: **teacher-generated data contains noise and inconsistencies, and distilling only responses causes the student to blindly imitate these errors.** To “distill not only data but also rewards,” we must transform the teacher’s noisy, query-specific evaluations into a stable and reusable reward function. This cannot be achieved by directly using the teacher online during RL.
>
> - **Teacher scores are inherently noisy and unpredictable across different queries; they do not provide the consistent supervision required for RL optimization.** A reward model, trained on aggregated teacher evaluations, serves to denoise, standardize, and stabilize these signals, enabling the student to learn what constitutes a good answer rather than simply copying teacher outputs.
> - Furthermore, our teachers are large, logit-free models (8B–70B or API-based), making **online querying during PPO prohibitively expensive.** Training a small reward model allows us to perform RL efficiently, **whereas direct preference KD and other logit-based approaches are not applicable in this black-box setting.**
>
> In short, a separate RM is essential not for methodological convenience, but because it is the only practical and principled way to (i) **correct teacher noise** and (ii) **support scalable RL in the logit-free distillation regime**.
>
> Additionally, in practice, **our RM training is not expensive:**
> * All data labeling is performed automatically by the teacher (no human annotation is required).
> * For a Llama 3–1B reward model on GSM8K, we use 1 H100 GPU; 4 epochs take about 1 hour.
> * For a 3B reward model, we use 4 H100 GPUs; 4 epochs also take about 1 hour.
>
> We added this discussion of computational cost to the Appendix.

---

> ### Author Response · Authors · 2025-11-18
>
> ## Response to Weakness 5:
> > W5: Unanalyzed Computational Cost: The paper ignores the massive computational cost of its pipeline. The reward generation requires 10 teacher samples + 30 student samples and evaluations per query. This cost is not analyzed, and it's plausible that a baseline SFT model trained with the same compute budget (i.e., 40x more data) would perform just as well.
>
> Thanks for pointing this out. We provide a concrete computational cost analysis on GSM8K below and include it in Appendix F:
> - Teacher generation + warm-up SFT. This does not incur additional cost beyond standard SFT, since it is a shared phase with conventional supervised distillation pipelines.
> - Student generation (for reward model training):
>     - Llama-1B: inference once over the selected data on 1 H100 takes about 6 min. For 30 answers, 6 min × 30 = 3 hours.
>     - Llama-3B: inference once on 1 H100 takes about 0.25 hours. For 30 answers, 0.25 hours × 30 = 7.5 hours.
> - Reward model training (distilling reward signals from the teacher’s responses):
>     - Llama-1B reward model: with 1 H100, 4 epochs cost 1 hour.
>     - Llama-3B reward model: with 4 H100, 4 epochs cost 4 hours.
> - PPO training:
>     - Llama-1B student: with 4 H100, about 8 hours for 5k steps.
>     - Llama-3B student: with 4 H100, about 32 hours for 8k steps.
>
> **Performance analysis of more SFT data:** We investigate this in Figure 11 in the appendix. As the SFT data ratio increases from 0.5× to 1.0×, performance steadily improves; however, further increasing it to 1.2× leads to a slight drop in accuracy. This indicates that using more SFT data is **not always beneficial**, likely because additional teacher-generated data introduces more noise and errors.
> This also supports our claim that the student should not only mimic the teacher's output but also learn to distinguish the good answers from the bad ones.
> As a result, we select the data ratio that achieves the best warm-up performance (1.0× in the figure) rather than blindly using more SFT data.
>
> ---
> We hope our explanations and proposed revisions address your concerns, and we kindly request you to reevaluate our work. We are truly thankful again for your time and efforts!

---

### Meta-Review · Area_Chair_w9TE · 2025-12-22

**Summary:**

The reviewers raise several concerns about the novelty, baselines, and practical significance of the proposed distillation framework. The core contribution relies primarily on a combination of existing techniques (e.g., majority voting, LLM-as-Judge, PPO with a separate reward model) without introducing significant methodological novelty. Moreover, the proposed three-stage pipeline introduces substantial additional training overhead compared to standard distillation. Furthermore, the empirical evaluation lacks comparison against more advanced distillation methods. Finally, the reliability of the distilled reward signal under noisy or failure-prone teacher evaluations is not adequately addressed, raising doubts about the robustness and general applicability of the approach. In light of these issues, I recommend rejection.

**Reviewer Concerns:**

Although the authors' rebuttal clarifies some key points (such as framing the problem as black-box, logit-free distillation, providing a cost analysis, and showing that simply scaling SFT data does not reliably improve performance), it only partially addresses the reviewers' core concerns. Several major issues remain unresolved: First, the claim that distilling a separate reward model denoises the teacher's noisy feedback lacks empirical support; the paper offers no ablation comparing a distilled RM against using the teacher's raw scores directly (e.g., in PPO with online LLM-as-Judge or via offline methods like DPO). Second, the necessity of distilling the teacher's own reward signal is questionable since high-quality external reward models or ground-truth labels are often available, cheaper, and more reliable than teacher-generated pseudo-labels, yet the paper fails to demonstrate that its distilled RM outperforms these alternatives. Consequently, the added complexity of jointly distilling data and reward from the same teacher is not sufficiently justified. Finally, the rebuttal does not adequately address concerns about the paper's limited novelty. As a result, while the rebuttal improves technical clarity, it does not substantively alter the overall assessment of the paper's contribution or quality.

**Reviewer Scores:**

I believe none of the reviewers would have changed their scores:

* Ko1H: The rebuttal clarified the black-box setting but did not overcome the core concern about limited novelty.
* X6QW: The authors provided no empirical comparison to relevant baselines; without such results, the claimed advantage of distilling teacher rewards remains unproven.
* 5QHe: The rebuttal addressed a fairness concern but did not enhance the paper's contribution enough to warrant a higher score.
* 5AyG: The response offered only theoretical justifications, with no experimental validation of reward reliability or robustness.

In summary, while the rebuttal improved clarity on several technical points, it did not sufficiently address the fundamental issues of novelty, baseline completeness, empirical validation, and robustness that underpin the reviewers' scores.

---

### Decision · Program_Chairs · 2026-01-26

Reject